**Article**  https://doi.org/10.1038/s41467-023-41640-9

# Structure of the peroxisomal Pex1/Pex6 ATPase complex bound to a substrate

Maximilian Rüttermann [1,2], Michelle Koci[3], Pascal Lill[1,2,3], Ermis Dionysios Geladas [1,2], Farnusch Kaschani [4], Björn Udo Klink [1,2], Ralf Erdmann [5] & Christos Gatsogiannis [1,2,3] ✉

The double-ring AAA+ ATPase Pex1/Pex6 is required for peroxisomal receptor recycling and is essential for peroxisome formation. Pex1/Pex6 mutations cause severe peroxisome associated developmental disorders. Despite its pathophysiological importance, mechanistic details of the heterohexamer are not yet available. Here, we report cryoEM structures of Pex1/Pex6 from *Saccharomyces cerevisiae*, with an endogenous protein substrate trapped in the central pore of the catalytically active second ring (D2). Pairs of Pex1/Pex6(D2) subdomains engage the substrate via a staircase of pore-1 loops with distinct properties. The first ring (D1) is catalytically inactive but undergoes significant conformational changes resulting in alternate widening and narrowing of its pore. These events are fueled by ATP hydrolysis in the D2 ring and disengagement of a "twin-seam" Pex1/Pex6(D2) heterodimer from the staircase. Mechanical forces are propagated in a unique manner along Pex1/Pex6 interfaces that are not available in homo-oligomeric AAA-ATPases. Our structural analysis reveals the mechanisms of how Pex1 and Pex6 coordinate to achieve substrate translocation.

Peroxisomes are single-membrane enclosed eukaryotic organelles playing central roles in lipid metabolism and maintenance of redox balance[1–3]. Impaired peroxisomal assembly and/or defects in reactive pathways housed by peroxisomes manifest in severe metabolic disorders[4–6]. There is also increasing evidence correlating peroxisomal dysfunction with aging and a broad range of relevant diseases, including cancer, Parkinson's disease and diabetes[7–9]. Peroxisomal enzymes are produced in the cytosol and then imported into the peroxisomal matrix[10]. The majority of peroxisomal matrix enzymes carry a C-terminal Peroxisome Targeting Signal (PTS1), which is recognized by the peroxisomal receptor Pex5 in the cytosol[11]. Pex5 delivers its cargo to a docking complex at the peroxisomal membrane[12,13]. This interaction triggers the formation of a transient nuclear-pore like assembly[14] for cargo translocation along the

peroxisomal membrane[15,16]. The underlying mechanism is not yet understood.

It is thought that during this process, the receptor either enters the membrane to become part of the channel (similar to a pore-forming toxin) or accompanies the cargo and enters completely into the peroxisomal lumen[16–18]. For the next round of import, the receptor has to be recycled and thus translocated back to the cytosol[17,19]. The peroxisomal ubiquitin-ligase complex Pex2/Pex10/Pex12 was recently shown to contain a pore that might provide the retro-translocation path for the receptor[20]. The unstructured N-terminal peptide of the receptor was suggested to enter this pore from the peroxisomal lumen, and subsequently gets mono-ubiquitylated by the ring-finger peroxin Pex2 of the ubiquitin ligase complex[20,21]. The Pex5 ubiquitin moiety is subsequently recognized by the type II (AAA)+

[1]Institute for Medical Physics and Biophysics, University Münster, Münster, Germany. [2]Center for Soft Nanoscience (SoN), University Münster, Münster, Germany. [3]Department of Structural Biochemistry, Max Planck Institute of Molecular Physiology, Dortmund, Germany. [4]Analytics Core Facility Essen, Center of Medical Biotechnology (ZMB), Faculty of Biology, University of Duisburg-Essen, Essen, Germany. [5]Institute for Biochemistry and Pathobiochemistry, Department of Systems Biochemistry, Ruhr-University Bochum, Bochum, Germany. ✉e-mail: christos.gatsogiannis@uni-muenster.de

heterohexameric complex Pex1/Pex6, which is attached to the peroxisomal membrane via the tail anchored protein Pex15 (Pex26 in mammals)[22–24]. Under ATP consumption, Pex1/Pex6 is expected to process Pex5 by pulling it through its central ATPase pore back to the cytosol[24,25]. Receptor recycling by Pex1/Pex6 consumes energy, but Pex5 import is ATP independent[26,27]. However, both machineries (cargo import and receptor export) are functionally linked via Pex15[28] and the ubiquitination of Pex5[19,29]. This supports the idea of an export-driven peroxisomal enzyme import, with Pex1/Pex6 being the driving molecular motor for a sustainable import[30–32]. When Pex1 and/or Pex6 are impaired, ubiquitinated Pex5 accumulates at the peroxisomal membrane, which triggers pexophagy[21,33]. More recently, the autophagy receptor Atg36 has been proposed as a Pex1/Pex6 substrate in yeast[33]. This suggests that Pex1/Pex6 might have additional functions in organelle quality control[34], which may explain the mostly lethal phenotype of peroxisomal biogenesis disorders (PBDs)[34].

Pex1 and Pex6 are both composed of two N-terminal domains (N1 and N2) followed by two AAA cassettes, known as the D1 and D2 domains (Fig. 1a). Each cassette is comprised of two distinct sub-domains: a core nucleotide-binding domain (large ATPase subdomain, $_L$D1/ $_L$D2) and a smaller domain of α-helical bundles (small ATPase domain, $_S$D1/ $_S$D2). The tandem N-terminal domains (NTDs) are a unique feature of Pex1 and Pex6[35]. Other members of the type II AAA-ATPase family, for example Cdc48/p97 or Rix7, contain only a single N-terminal domain, which is nevertheless structurally related to the N-terminal domains of Pex1/Pex6 and known to mediate a plethora of interactions with co-factors and adaptor proteins[36]. The D1 and D2 domains form two stacked heterohexameric rings with a central channel, but in contrast to other type II AAA-ATPases, only the D2 domains of Pex1 and Pex6 are capable to hydrolyze ATP[35,37,38]. Both Pex1 and Pex6 are required for assembly, which occurs in an ATP-dependent manner[39,40].

In contrast to the D1 and D2 domains, the N-terminal domains of Pex1 and Pex6 do not display high sequence identity to each other and might be responsible for the specific functions of Pex1 and Pex6. For example, the membrane anchor of Pex1/Pex6 (ScPex15, HsPex26) binds exclusively to Pex6[22], whereas the autophagy receptor Atg36 (yeast) binds exclusively to Pex1[33]. The N-terminal regions are not well characterized and might perform additional functions, for example binding to the ubiquitin moiety of Pex5[41] or association with the ubiquitin hydrolase Ubp15p[42].

Previous negative stain EM studies in different nucleotide states revealed the general architecture of the heterohexameric Pex1/Pex6 double ring complex, consisting of alternating subunits of Pex1 and Pex6, and suggested large conformational changes in the D1 and D2 rings[24,35,38]. In addition, cryo-EM structures of Pex1/Pex6 in the presence of ADP and ATPγS were determined with an overall resolution between 6.2 Å and 8.8 Å[37]. In contrast to the negative stain EM data, the available cryo-EM structures did not indicate drastic conformational changes and opening of the pore between the ATPγS and ADP states[37]. Despite its essential role in peroxisome biogenesis, import of matrix enzymes and quality control[43], detailed structural insights into Pex1/Pex6 architecture and mechanism of substrate processing are not yet available.

Numerous high-resolution structures of double-ring ATPases in complex with substrates have now been solved, providing important insights into a rather conserved mechanism of substrate processing through the central pore formed by the D1 and D2 domains[44–50]. However, the coordination between the D1 and D2 ATPase domains in Pex1/Pex6 is complex, involves two different proteins and communication "hubs" between their inactive D1 and active D2 domains. Moreover, it is still unclear whether the underlying molecular mechanisms involve conformational changes of the unique tandem N-terminal domains. For a comprehensive understanding of Pex1/Pex6 function, it is crucial and of considerable biological and pathophysiological interest to visualize Pex1/Pex6 in a "working", substrate-bound state.

In this study, we capture Pex1/Pex6 from *S. cerevisiae* with an endogenous substrate in the central pore and determine two Pex1/Pex6 cryo-EM structures in different states. These structures reveal the complex coordinated interplay between Pex1 and Pex6 during substrate processing and allow us to highlight unique features of the molecular motor of peroxisomal receptor recycling.

## Results

In order to obtain high resolution structural information for Pex1/Pex6, we first introduced the E832Q mutation to the Walker B ATPase motif of the D2 domain of Pex6 from *S. cerevisiae* (Pex6_WB). As reported previously, the ATPase activity in the Pex1/Pex6_WB is heavily affected, despite the presence of WT Pex1[24,35,37,38] but not completely depleted (Supplementary Fig. 1). Considering that the D2 domains are responsible for all ATP hydrolysis in Pex1/Pex6[35], this suggests that the D2 domains of Pex1 and Pex6 control the ATPase activity of the complex in a distinct and possibly highly coordinated manner.

We co-expressed Pex1 WT and Pex6_WB in *S. cerevisiae* and purified the complex by affinity chromatography and size-exclusion chromatography in the presence of a saturating concentration of ATP (1–3 mM) (Supplementary Fig. 2a, b). We utilized single particle cryo-EM analysis (Supplementary Fig. 2c, d) and finally obtained cryo-EM maps of two distinct states of the complex (Supplementary Fig. 3).

The better resolved conformation (Fig. 1, Class 3 in Supplementary Fig. 3; Supplementary Movie 1) shows Pex1/Pex6_WB in an overall closed conformation with a well-resolved symmetric D1 ring (average resolution 3.7 Å (FSC = 0.143); 3.3 Å (FSC = 0.5) upon density modification) (Supplementary Fig. 4). The asymmetric D2 ring is resolved to 3.9 Å (3.6 Å (FSC = 0.5) upon density modification) (Supplementary Fig. 3). The particle shows a characteristic triangular shape of alternating Pex1 and Pex6 subunits that are arranged around the central ATPase pore. The D1 ring is stacked on top of the D2 ring and crowned by the N-terminal domains (NTDs) (Fig. 1b–d). The D2 ring displays a well-resolved density for five of the ATPase domains and a partially fragmented density for the remaining ATPase domain (4.5 to 7 Å (FSC = 0.143), 4 to 5.5 Å (FSC = 0.5) upon density modification) (Supplementary Fig. 4). The NTDs of Pex1 and Pex6_WB are well resolved (Fig. 1b), except the N1 domain of Pex1, which is flexibly attached to the rest of the protein via a long linker peptide of 15-20 residues (Fig. 1a, c).

We derived a molecular model of this conformation (Supplementary Fig. 5). Surprisingly, during modeling, we observed a clear density for a peptide of ~9 residues passing through the central pore of the D2 ring (Fig. 1b; inset). This result was rather unexpected, as we did not add any substrate during sample preparation, nor did we establish growth conditions under which peroxisomal substrates might be present in a stoichiometry to the recombinant complex. When Pex6 D2 carries a WB mutation, the complex maintains a residual basal ATPase activity (Supplementary Fig. 1). The attenuated hydrolytic activity has proven to be advantageous in "trapping" the AAA+ motor in a substrate-engaged state for our structural studies similar to other AAA-ATPases[50–52]. The density is, however, not sufficiently resolved to allow a clear identification of the polypeptide. Qualitative mass spectrometry analysis of the recombinantly expressed Pex1/Pex6 complex did not allow us to unambiguously identify the substrate, as all peptides were non-peroxisomal and could therefore be non-specific substrates. Interestingly, known substrates such as Pex5[19,53], Pex15[24] of Atg36[33] were not present in our preparation. It is important to note that the complex was overexpressed under galactose-induced conditions, which may lead to an imbalance in the stoichiometry between peroxisomal proteins and the recombinant complex. A true substrate should ideally be present in a 3:1 stoichiometry with both Pex1 and Pex6. The density probably corresponds either to a mixture of several

endogenous substrates trapped in the D2 pore during purification, or to an event of self-unfolding of the complex, as was recently observed in the cryoEM structures of Rix7[49] and VAT[54].

Furthermore, we were not able to identify substrate density in or above the D1 channel or below the D2 ring. This is consistent with cryo-EM structures of p97 bound to small substrates (Ub₆), where substrate density was also only resolved in the D2 ring[44].

### N2 and D1 domains mediate Pex1/Pex6 complex assembly

In this conformation (Fig. 1), the D1 ring is symmetric and all ATPase domains of the D1 ring are bound to ATP (Fig. 2a–c; Supplementary Fig. 6a; Supplementary Fig. 7a–c). Nucleotides bind in a pocket formed

at the interface between the large (LD1) and small (SD1) subdomains of Pex1(D1) and the large subdomain (LD1) of Pex6(D1) and vice versa (Fig. 2a).

The NTDs of Pex1 and Pex6 differ significantly in their conformation relative to the D1 ring (Fig. 1b–d). The N2 domains of Pex1 and Pex6 are closely associated and form a third layer of three Pex6(N2)-Pex1(N2) dimers that crown the D1 ring (Figs. 1c, d and 2b). The flexible Pex1(N1) domain, that is not resolved in our cryo-EM density, is probably located above the N2 layer (Fig. 1c, d). According to previous low resolution EM studies, Pex1(N1) adopts an extreme-"up" conformation (see[38]). In contrast, Pex6(N1) is well resolved and adopts a characteristic "down" conformation that is coplanar with the

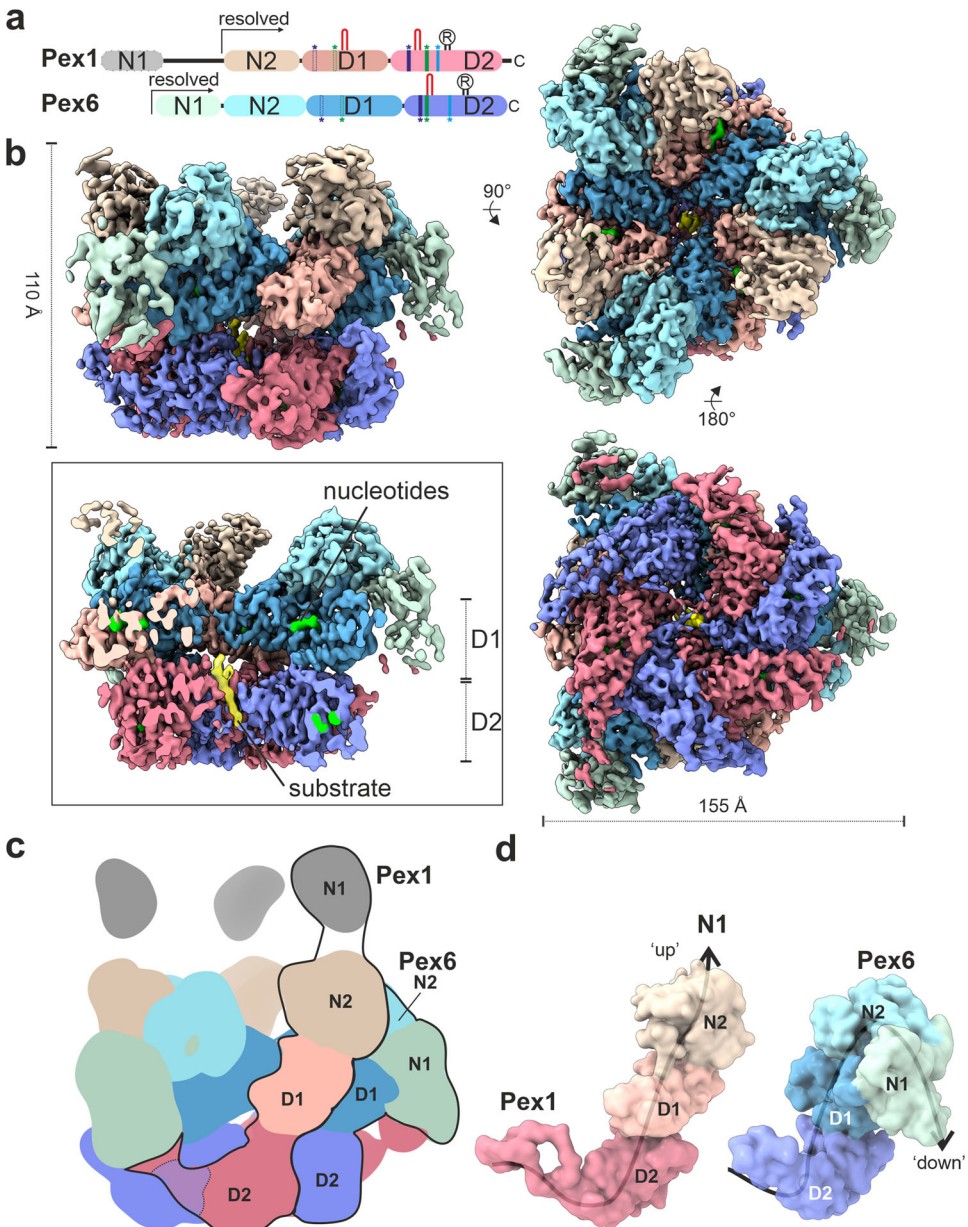

**Fig. 1 | Cryo-EM structure of the peroxisomal ATPase Pex1/Pex6. a** Schematic representation of the Pex1 and Pex6 primary structures colored by domain. The color code is maintained throughout the manuscript. Important conserved motifs are indicated: Walker A (dark purple), Walker B (green), inter-subunit signaling (ISS) (cyan), pore loops (red loop), and arginine fingers (R). Dashed boxes (D1) indicate degenerated motifs. **b** Cryo-EM density map of the better resolved 3D class 3 (Supplementary Figs. 3, 4; termed "single-seam" throughout the manuscript) from yeast Pex1-Strep₂/His-Pex6^(E832Q) with each subunit and domain colored as in

**a** shown as side, top and bottom view. The inset shows a cut-away view of the cryo-EM structure displaying the central ATPase channel. The density of the substrate and nucleotides are shown in yellow and green, respectively. Density-modified map is shown at 2.26 sigma. **c** Cartoon depicting the overall domain organization of Pex1 and Pex6. The flexible Pex1(N1) domain is not resolved in our cryo-EM density, but shown in an "up" conformation, in accordance with previous low resolution EM studies[38]. **d** Surfaces of the molecular models of Pex1 and Pex6 are shown in the same orientation. Note the characteristic "down"-conformation of Pex6(N1) (right).

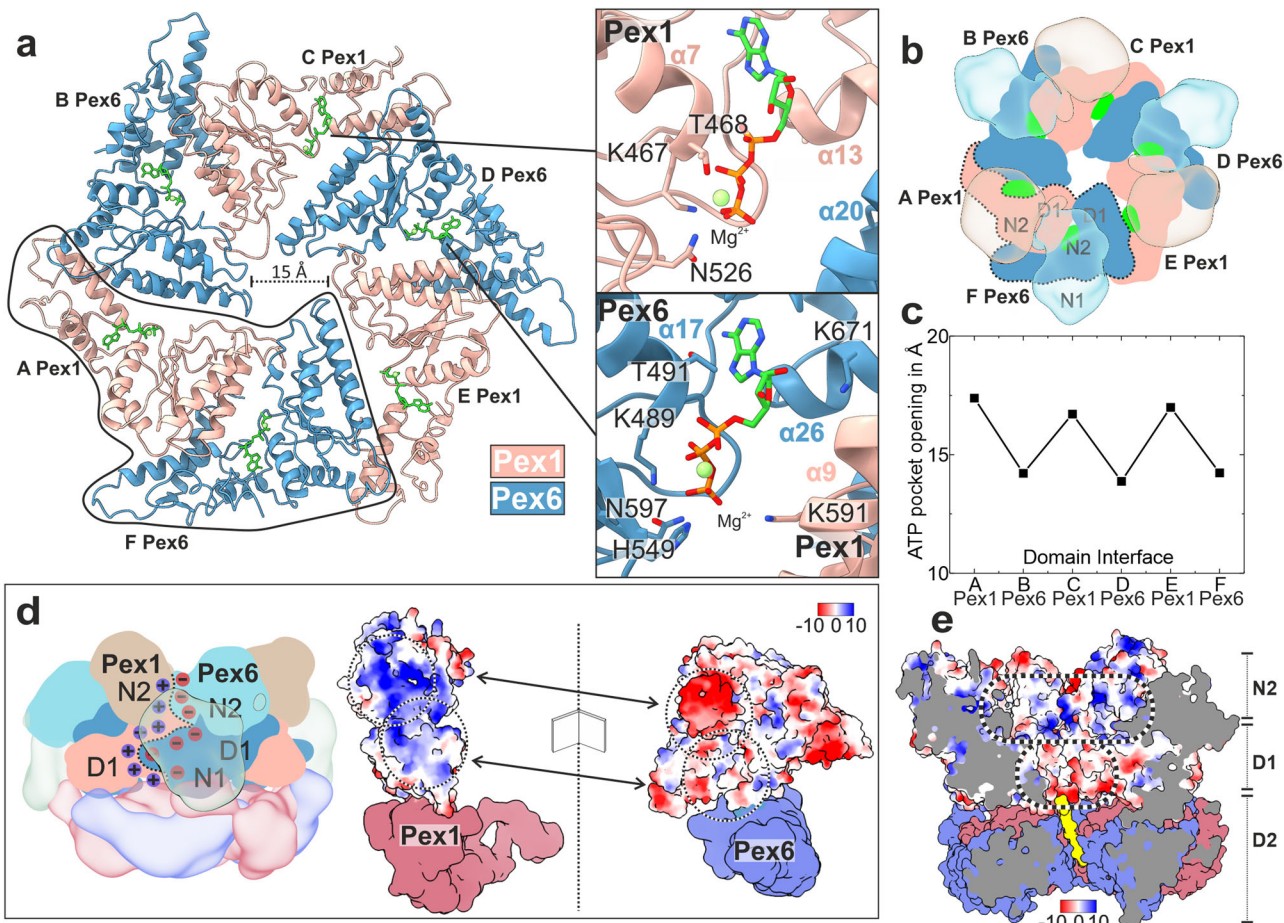

**Fig. 2 | Arrangement of D1 ring and the N-terminal domains. a** Top-view of the molecular model of the D1 ring, which resembles a trimer of "tight" Pex6(D1)-Pex1(D1) dimers. Pex1(D1) (beige), Pex6(D1) (cyan), ATP (green). The insets show magnified images of the ATP binding sites at the intra-dimeric Pex6(D1)-Pex1(D1) (upper image) and inter-dimeric Pex1(D1)-Pex6(D1) interface (lower image). **b** Top-view of the symmetric D1 ring (solid colors) capped by the N-terminal domains (transparent) shown as cartoon. **c** Plot of the nucleotide binding pocket opening of both interfaces measured as distances between the Cα atom of Pex1 T468 to the clockwise-neighboring Pex6 D582 and Pex6 T491 to Pex1 K563, respectively. **d** Side view cartoon representation of Pex1/Pex6. The interface between Pex6(N2,D1)/Pex1(N2,D1) dimers (clockwise) is characterized by strong electrostatic interactions. The interface is flipped open to demonstrate the complementary charges at the surface (electrostatic Coulomb potential at pH 7.5). Scalebar kcal/(mol·e) at 298 K. **e** Surface electrostatic Coulomb potential of the N2-D1 layers at pH 7.5 Positively- and negatively charged surfaces are colored in blue and red, respectively. Scalebar kcal/(mol·e) at 298 K.

D1 ring (Fig. 1c, d). The flexible Pex1(N1) that is not resolved in our cryo-EM structure might play a role in substrate recognition and adopt distinct conformations upon interaction with yet unknown adaptor proteins.

The Pex1(N2) and Pex6(N2) domains are involved in strong interactions to each other and the D1 ring, rendering large independent conformational changes of these domains rather unlikely. In particular, Pex1(N2) and Pex6(N2) form "tight" N2-dimers via strong complementary electrostatic interactions (Fig. 2d). The symmetric D1 ring resembles in general a "trimer" of Pex6(D1)/Pex1(D1) dimers arranged in a clockwise-manner (when viewed from D1 towards D2: top view) (Fig. 2a, b). Each Pex6(D1)/Pex1(D1) dimer is also capped by the "tight" (N2)-dimer, which contributes complementary charges that significantly enforce this interface (Fig. 2d). The intra-dimeric (Pex6(D1)/Pex1(D1)) interface is additionally stabilized by ATP (Fig. 2a). Pex6 provides a degenerated Walker-A motif (T491 and K489), a histidine (H549) and a lysine (K671) whereas Pex1 provides an additional lysine (K591) (degenerated Arg-finger) to the heteromeric binding pocket of the nucleotide (Fig. 2a; lower inset). Although the Walker A motif of Pex6 is not archetypal, Pex6(D1) can still bind ATP, which can however not be further hydrolyzed, due to substitutions in the Walker-B and R-finger motifs (Fig. 2a; lower inset).

The Pex1(D1)-Pex6(D1) interface between dimers is less compact and does not involve N-terminal domains (Fig. 2b). This is reflected in the larger opening of the nucleotide binding pocket between Pex1(D1) and Pex6(D1) (Fig. 2a, upper inset), when compared to the respective pocket at the intra-dimeric interface (Fig. 2a, lower inset). In this case, ATP interacts with residues T468 and K467 (Walker A-like), and N526 (Walker B-like) that are contributed by Pex1(D1). Surprisingly, Pex6(D1) does not interact with the ATP bound to Pex1, due to the large distance of the involved domains and the deletion of the R-finger motif (Figs. 1a and 2a). Thus, ATP does not directly contribute to the interdimeric contact, but previous studies report that mutation of the conserved Walker A lysine 467 within the D1 domain of Pex1 results in an assembly defect[38,55]. This suggests that although ATP at Pex1-D1 does not directly link the adjacent subunits, it may instead stabilize the small ATPase domain of Pex1, which is directly involved in the interdimer interaction.

Despite high structural similarity, Pex1(SD1) and Pex6(SD1) differ significantly in size (Supplementary Fig. 8a, b). Pex6(SD1) is 20 residues longer than Pex1(SD1) (Supplementary Fig. 8b; Pex6 helix α26 and α27). This feature dictates the compactness of the interface between SD1 and LD1. The loose interface between dimers (Pex1(SD1)→Pex6(LD1)) involves the "shorter" Pex1(SD1), whereas the compact dimeric interface (Pex6(SD1)→Pex1(LD1)) involves the "longer" Pex6(SD1) (Fig. 2a;

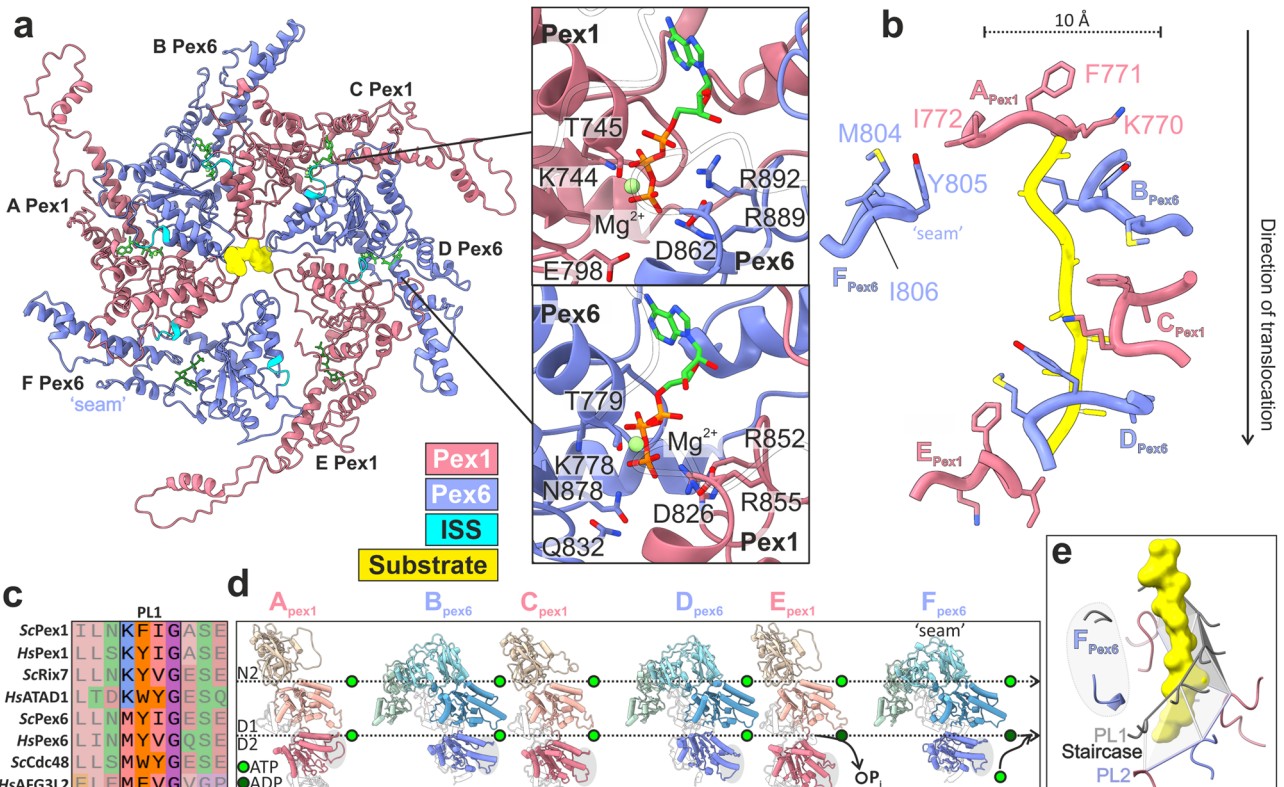

**Fig. 3 | Substrate engagement in the active Pex1/Pex6 D2 channel. a** Top view of the substrate-bound Pex1/Pex6 D2 ring. Pex1(D1) (coral), Pex6(D2) purple, ISS motif (cyan), nucleotides (green) and substrate (yellow). The insets show magnified images of the ATP binding sites at the Pex1(D2)-Pex6(D2) (upper image) and Pex6 (D2)-Pex1(D2) interface (lower image) of substrate engaged subunits C and D. **b** Residues of pore loops 1 contacting the substrate. **c** Multiple alignments of pore loop 1 sequences of D2 domains from several AAA-ATPases including Pex1, Pex6, Rix7, p97/Cdc48, AFG3L2 and ATAD1. **d** Pex1 and Pex6 subunits shown side-by-side, aligned along their D1 domain. Note the sequential tilt of the D2 domain relative to the D1, which is highlighted by a dashed line above each domain. The catalytically "dead" D1 domains remain ATP-bound. **e** Spiral arrangement of pore loops 2. Pore loops 1 are shown in grey and substrate in yellow. The pore loops 2 of subunits Pex1(A) to Pex6(E) (PL2) form a second spiral staircase running below and parallel to the spiral staircase formed by the pore loops 1 (PL1). The pore loops of the "seam" subunit Pex6(F) (highlighted in gray) are displaced significantly from the spiral staircase.

Supplementary Fig. 8a). In Pex6($_S$D1)→Pex1($_L$D1) (compact dimeric interface; (Fig. 2a; Supplementary Fig. 8a), M679 of the prolonged helix α26 of Pex6($_S$D1) docks into a hydrophobic pocket presented by Pex1($_L$D1) (Supplementary Fig. 8c). Furthermore, the prolonged helix α26 of Pex6($_S$D1) displays a negatively charged pocket that allows binding of the conserved H592 of Pex1($_L$D1) (Supplementary Fig. 8c). In contrast, at the less compact Pex1($_S$D1)→Pex6($_L$D1) interface between dimers, the analogous H609 of Pex6($_L$D1) is not involved in any interactions (Fig. 2a; Supplementary Fig. 8d, yellow highlight), due to the deletion in the opposite helix α14 of Pex1($_S$D1). This allows the analogous hydrophobic pocket of Pex1($_L$D1) to click further downstream into the conserved Y654 of the shorter helix α13 of Pex1($_S$D1) (Supplementary Fig. 8d). The consequence is a larger angle between Pex1($_S$D1) and Pex6($_L$D1) and thus a less compact interface between two adjacent dimers. In summary, the deletion in helix α13 and α14 of Pex1($_S$D1) determines the "trimer-of-dimers" arrangement in the D1 ring of Pex1/Pex6 complex.

Each D1 heterodimer is further capped by the associated NTDs, which further results in the characteristic triangular shape of Pex1/Pex6 when viewed from the top (Figs. 1b, 2b). The resulting central pore of the D1 ring has a diameter of approximately 15 Å (Fig. 2a; Supplementary Fig. 7d). The D1 channel, however, lacks substrate density (Fig. 1b). This renders tight interactions between D1 and the substrate unlikely. Interestingly, the "crown" assembled by three N2 heterodimers forms a positively charged funnel leading to the negatively charged "mouth" of the D1 ring (Fig. 2e).

## The Pex1/Pex6 D2 ring motor processing substrate

Instead of a planar pseudo-symmetric arrangement, as observed in the previous reconstructions of Pex1/Pex6 in the presence of ADP and ATPγS[37], the AAA-ATPase domains of the D2 ring of substrate bound Pex1/Pex6 assemble into an asymmetric right-handed spiral staircase (Fig. 3), similar to other ATPases[56,57]. The substrate polypeptide chain presumably stabilized this conformation. Similar to the D1 ring, Pex1(D2) and Pex6(D2) are also organized in pairs (Fig. 3a). The Pex6($_S$D2)→ Pex1($_L$D2) dimeric interface is indeed stronger associated than the Pex1($_S$D2)→ Pex6($_L$D2) interface between the D2 dimers (Supplementary Fig. 9).

The D1 and D2 rings are rather flexibly connected via the D1→D2 linker peptides and the protrusion domains of Pex1(D2)(helix α28) and Pex6(D2)(helix α38) (Supplementary Fig. 10a, b). The protrusion domain of Pex6(D2) interacts with Pex1(D1) within a subunit dimer (Supplementary Fig. 10c, lower panel), whereas the more flexible protrusion domain of Pex1(D2) (Supplementary Fig. 10b, c) interacts with the neighboring Pex6(N1), linking thereby two adjacent subunit dimers (Supplementary Fig. 10c, upper panel).

Within the D2 ring, the interfaces between the D2 subdomains (Pex1(D2)/Pex6(D2) (between dimers) and Pex6(D2)/Pex1(D2) (intra-dimer)) are further stabilized by nucleotides. The binding pockets are formed by conserved Walker A and Walker B residues interacting with a pair of arginine-finger residues of the clockwise neighboring subunit (Pex1: R852, R855; Pex6: R889, R892) (Fig. 3a; Supplementary Fig. 6a, lower panels). Mutation of the Arg-finger residues Pex1(R852) and

Pex6(R889) to lysine results in 100% and 80% inhibitions of the ATPase activity of the complex, respectively[35,38].

The substrate is threaded into a right-handed spiral staircase formed by pore loops 1 of Pex1(D2) and Pex6(D2). The central pore of the spiral staircase has a diameter of ~10 Å (Fig. 3a, b). The density of the substrate is ambiguous and we therefore modeled a continuous poly-alanine polypeptide backbone of 9 amino acids with N → C directionality (Fig. 3b). This is consistent with the translocation direction of other substrate-engaged cryo-EM structures of type II ATPases processing ubiquitinated substrates[44,45,47], as suggested for Pex1/Pex6[25,41,53], and exhibiting the same characteristic right-handed spiral staircase arrangement of pore loops 1, as observed in our structure. The pore loops 1 of chain A (Pex1) and E (Pex1) occupy the highest and the lowest position of the spiral (Fig. 3b, grey highlight in 3d). D2 of chain F (Pex6) does not directly contact the substrate and does not follow the spiral arrangement ("seam" subunit) (Fig. 3a, b). We therefore define this conformation of the heterohexamer as the "single-seam" state. The substrate engaged pore loops 1 of D2 domains Pex1(A,C,E) and Pex6(B,D) are related to each other by a ~60° rotation (axis of Pex1-helix $\alpha_{22}$ to Pex6-helix $\alpha_{30}$) and a distance of 8–9 Å (Pex1 residue 771 Cα to Pex6 residue 805 Cα). The D2 pore loops 1 of Pex1 and Pex6 contain only a single aromatic residue in a **KFI**- or **MYI**- motif, respectively. The aromatic residue (Pex1 F771; Pex6 Y805) and the isoleucine of the involved pore loops 1 bind to the substrate every two amino acids in a "forceps"-like manner (Fig. 3b). The preceding methionine (M804) of Pex6(D2) and lysine (K770) of Pex1(D2) mediate the communication between the adjacent pore loops and stabilize their arrangement (Fig. 3b). M804 of Pex6 contacts F771 of the adjacent clockwise Pex1(D2), whereas K770 of Pex1 forms π-cation interactions with the aromatic Y805 of the adjacent clockwise Pex6(D2) (Fig. 3b). The residue prior to the aromatic in pore loop 1 is in general suggested to have a significant role on substrate translocation[50]. The Pex1(D2) pore loop 1, with lysine forming π-cation interactions, is similar to the staircase of Rix7 and ATAD1 (Fig. 3c). The M804 of the Pex6(D2) pore loop 1 is rather engaged in van-der-Waals interactions, similar for example to Cdc48 or AFG3L2 (Fig. 3c). The D2 ring thus features two types of pore loops 1 with distinct characteristics. Whether and how these hybrid features of the spiral-staircase have an effect on fine-tuning substrate translocation and Pex1-Pex6 coordination during this process requires further investigation.

The four D2 domains at the top of the spiral are ATP-bound (Pex1(A), Pex6(B), Pex1(C), Pex6(D)) (Fig. 3b, d; Supplementary Fig. 6a; lower insets). Estimation of the nucleotide in the binding pocket of the 'bottom' Pex1(D) and 'seam' subunit Pex1(F) (Fig. 3b, d) based on cryo-EM density alone was not possible with absolute confidence (Supplementary Fig. 6a). The large number of available AAA-ATPase cryoEM structures with ADP bound to one or more binding pockets (PDB: 7SWL, 7T0V, 6AZ0, 7UQI, 7PX9, 7UPT, 6W22, 6W23, 6W24, 6P07, 7ABR, 7TDO, and 6SH3) show however a consensus regarding the position of the respective pore loops 1 in the spiral staircase (bottom or disengaged) and the opening of the nucleotide binding pocket. The extent of nucleotide binding pocket opening has been thus established as a reliable indicator of the nucleotide status of the respective subunit[46,50,58]. We therefore measured the buried surface area between adjacent protomers and distances between Walker A and arginine-fingers of the associated protomers (Supplementary Fig. 6c–e). In line with these observations, our analysis suggests that both the "bottom" Pex1(D) and the disengaged "seam" Pex1(F) D2 domains are indeed in an ADP-bound state (Fig. 3c, e, Supplementary Fig. 6a, c–e)[44,47,59].

The pore loops 2 of engaged Pex1(D2) and Pex6(D2), except pore loop 2 of "seam" Pex6(F)), surround the substrate and form a second spiral staircase, below the spiral staircase formed by the pore loops 1 (Fig. 3e, Supplementary Fig. 11a). The pore loops 2 are less well resolved, lack aromatic residues and are not tightly associated with the substrate. Similar to other AAA-ATPases[56], they provide polar and

negatively charged residues facing the substrate. This might support the translocation process by stabilizing the backbone of the substrate and preventing its refolding during threading.

## Pex1 and Pex6 are highly coordinated

The second cryo-EM structure of substrate-bound Pex1/Pex6_WB (Supplementary Fig. 12) shows a dynamic state where both the D1 and D2 rings of Pex1/Pex6 have an asymmetric configuration (Fig. 4a). One of the Pex6(D1)/Pex1(D1) dimers thereby swings out. This results in an open central D1 channel with a diameter of 20 Å, approximately 5 Å larger than the symmetric D1 ring in the "single-seam" state (Fig. 4a; Supplementary Fig. 7a, d–f). Interestingly, the outward Pex6(D1)/Pex1(D1) dimer is stacked on top of a compact characteristic pair of dislocated seam domains of the D2 ring Pex6(D2)(E)/Pex1(D2)(F) (Fig. 4a, b). We therefore term this conformation as the "twin-seam" state of the complex.

Note that all Pex1(D1) and Pex6(D1) domains of the D1 ring remain bound to ATP (Supplementary Fig. 6b; upper panels), but the deletion of the R-finger motif in Pex6(D1) apparently renders the inter-dimeric interfaces flexible. This allows even greater distances between the outward dimer and adjacent subunits (Supplementary Fig. 7a–c). Substrate density is again limited to the D2 channel (Fig. 4c, Supplementary Fig. 11b). In the "twin-seam" state, only four alternating Pex6(D2) and Pex1(D2) domains (Pex6(A), Pex1(B), Pex6(C), Pex1(D)) form the substrate-interacting spiral staircase of pore loops 1 (Fig. 4c, Supplementary Fig. 11b).

The "twin-seam" dimer Pex6(E)/Pex1(F) is dislocated from the central channel (Fig. 4a). The pore loops 1 of the "seam" subunits are displaced significantly from the spiral staircase (Fig. 4c). This characteristic "twin-seam" pair is positioned between the lowest (Pex1(D)) and the highest position of the staircase (Pex6(A)) (Fig. 4c).

The loose interfaces of the "twin-seam" dimer to the neighboring D2 domains are profoundly different from the close contacts between the other D2 domains, including a rather "tight" interface between the two D2 domains within the "twin-seam" dimer (Fig. 4a; Supplementary Fig. 6c–e). The nucleotide binding site shared by the Pex1 (D2) "seam" (F) and the clockwise (from top) Pex6(D2) (A) is not well defined, thus it is either apo or ADP-bound (Supplementary Fig. 6b; lower panels (F); Supplementary Fig. 6c–e). Subsequent subunits contacting the substrate (Supplementary Fig. 6b; lower panels, (A,B,C), Supplementary Fig. 6c–e) exhibit strong density in the nucleotide binding pocket and tight interfaces between the respective motifs, suggesting that they are bound to ATP. The nucleotide binding site between the Pex6(D2) "seam" and the counter-clockwise Pex1(D2) (Supplementary Fig. 6b lower panels, (D); Supplementary Fig. 6c), show density suggestive of the presence of a nucleotide. The wide spacing between the involved motifs suggests that ADP is bound to this pocket (Fig. 4a; Supplementary Fig. 6c–e). The lowermost D2 domain of the staircase is thus bound to ADP and the "twin-seam" dimer is released from the staircase (Fig. 4c). Intriguingly, our analysis suggests that the nucleotide binding pocket (E) between the two "seam" domains remains bound to ATP (Supplementary Fig. 6b–e).

Taking into account that ATP-hydrolysis occurs in an anti-clockwise manner (when viewed from top), we compared the "twin"- (Fig. 4a) and the "single"- seam (Fig. 1b) structures and visualized the direction of motion between both states (Fig. 4b). It is interesting to note that both Pex1(D2) and Pex6(D2) of the "twin-seam" dimer move synchronously as a unit towards the preceding Pex6(D2) indicated as arrows (Fig. 4b). We assume that this drastic conformational change towards the "single-seam" conformation is triggered upon binding of ATP to the binding pocket (F) of the "twin-seam" (Supplementary Fig. 6c; lower panel). Parallel to the clockwise rotation (viewed from top) of the "twin-seam," (Fig. 4b, middle panel) the respective Pex6(N2/D1)/Pex1(N2/D1)-dimer swings in toward the central channel (Fig. 4b, upper panel), resulting in the symmetric D1 ring of the "single seam"

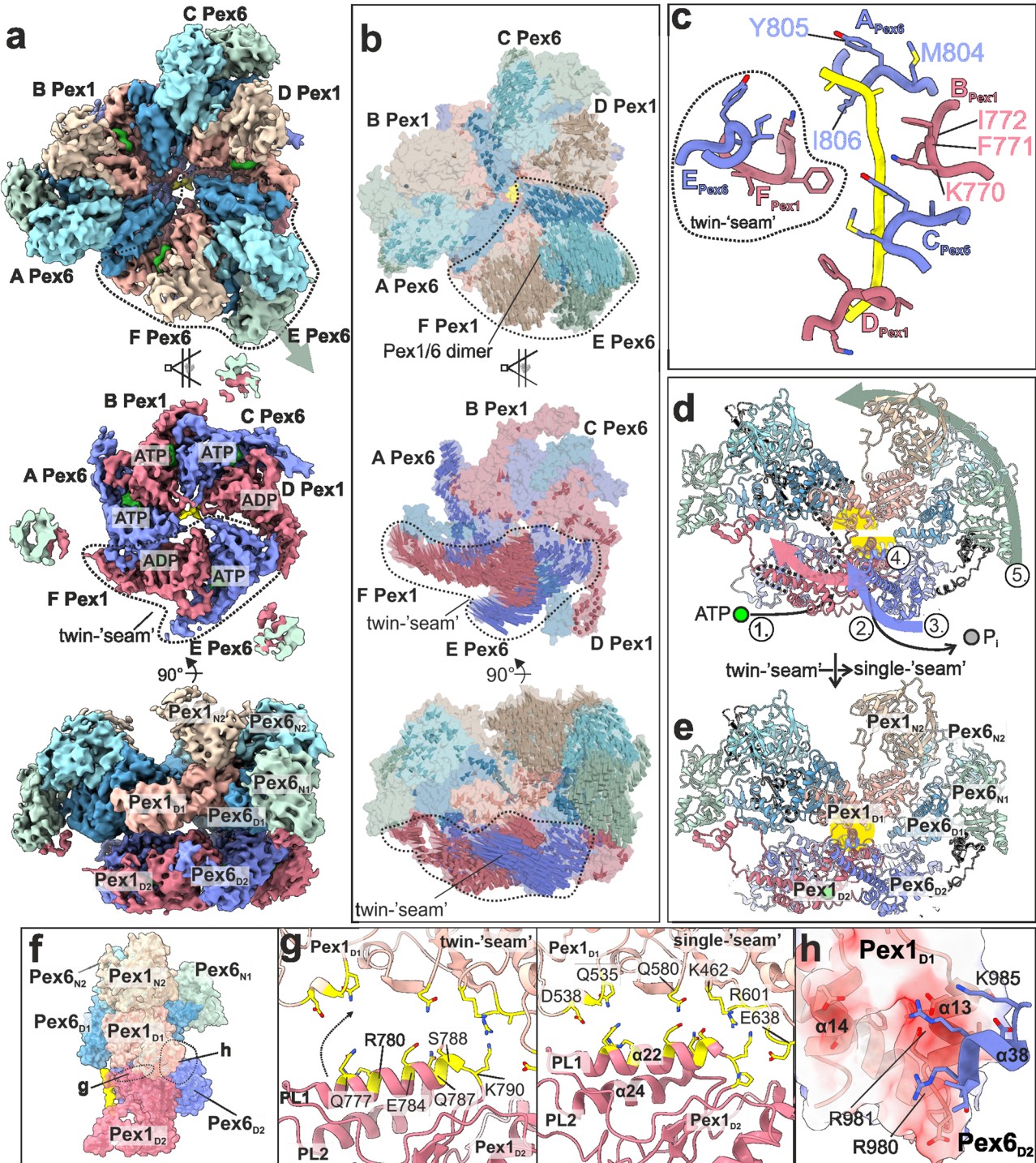

**Fig. 4 | Architecture of Pex1/Pex6 "twin-seam" state. a** Cryo-EM structure of "twin-seam" state of yeast Pex1/Pex6_WB shown as top (upper panel), cut- (middle panel) and side view (lower panel). Individual domains and the substrate are colored as in Fig. 1a. Note that the "twin-seam" subunits (dashed line) are dislocated from the substrate. Density-modified map is shown at 2.11 sigma. **b** Molecular surface of Pex1/Pex6 with vectors indicating the putative direction of motion during the conformational switch from the "twin-seam" to the "single-seam" state. **c** The pore loop 1 staircase with the twin-seam subunit being disengaged from the substrate. Residues of an individual loop of Pex1 and Pex6 are labeled. **d, e** Side view of the molecular model of Pex1/Pex6 dimer in the "twin-seam" **d** and "single-seam" **e** state. Possible sequential steps of the conformational switch are indicated: (1)

ATP-binding between dimers, (2) ATP-hydrolysis within the "twin-seam" dimer, (3) rotation, (4) D2 → D1 docking, and (5) D1, N swinging. The "communication hub" between the D2 and D1 ring is highlighted in yellow. **f** Side view of the Pex1/Pex6 dimer with interaction hubs being highlighted (Pex1(D2)/Pex1(D1) **(g)**; Pex6(D2)/Pex1(D1) **(h)**. **g** Pex1(D2)/Pex1(D1) interface in "twin-seam" (left panel) and "single-seam" state (right panel). This interface involves the Pex1(D2) helix α22 located downstream of pore loop 1. During the twin-seam (left panel) to the single-seam (right panel) transition, the helix moves, establishing new interactions. Residues are shown as sticks. **h** Pex6(D2)/Pex1(D1) interface (single-seam state). Surface of the negatively charged groove of Pex1(D1) colored according to the electrostatic potential. Interacting residues of Pex6(D2) are shown as sticks.

structure (Fig. 1b). Thus, there is a coordinated communication between the two rings, although the D1 ring is catalytically dead with respect to ATP hydrolysis.

Upon careful inspection of the conformational changes between the two states, we identified a major "communication-hub" between the D2 domain of Pex6 and the D1 domain of Pex1, within a Pex1/Pex6 dimer (Fig. 4d, e). Binding of ATP to the binding pocket of (F) triggers the rotation of the "twin-seam", so that "seam" Pex1(D2)(F) engages the substrate at the highest position of the spiral (Fig. 4a–e). This results in binding of pore loop 1 of Pex1(D2) (F) to the substrate, upwards movement of Pex1-helix$_{α22}$ (downstream helix of pore loop 1), which in turn establishes new strong interactions, pulling on Pex1($_L$D1) (Fig. 4d–g, Supplementary Movie 2,3). In parallel, the rotation of the Pex6(D2)/Pex1(D2) "twin-seam" dimer (Fig. 4b, middle panel, Supplementary Movie 2), pushes the Pex1($_S$D1) upward. This is mediated via the Pex6 protrusion helix α38 (Fig. 4d, f, h; Supplementary Fig. 10c(ii)). In particular, three positively charged residues of Pex6(D2)-helix$_{α38}$ (R980, R981, K985) dock into this groove and push Pex1($_S$D1) upward (Fig. 4h, Supplementary Movie 2; Supplementary Fig. 10c(ii)).

These conformational changes are propagated to the Pex6(D1)/Pex1(/D1) dimer on top of the "twin-seam", that swings toward the central channel and pulls the associated Pex6(N2) domain along that direction (Fig. 4b, d, e). However, it should be noted that the Pex6(N2) thereby retains its "down" orientation and remains connected to the Pex1(D2) protrusion domain. In summary, the described Pex6(D2)/Pex1(D1) pivot (Fig. 4d, e, yellow highlight) translates the rotation of the D2 "twin-seam" during the ATP hydrolysis cycle (Fig. 4b, middle panel) into a coordinated large inward tilting of the associated D1 dimer and N-terminal domains (Fig. 4b, upper panel).

## Discussion

Cryo-EM structures of numerous unfoldases revealed a highly conserved hand-over-hand mechanism of processive threading of diverse substrates along the central channel of AAA+ family members[44–52,57,60]. Pex1/Pex6 is a less well characterized type II AAA-ATPase that plays a crucial role in recycling of peroxisomal receptors. About 65% of PBDs (including Zellweger) are related to mutations of the human Pex1/Pex6 complex[4,5,33,34,61,62].

Pex1/Pex6 is the only known double-ring heterohexamer of the type II AAA+ family, with both subunits capped by a unique pair of N-terminal domains (Fig. 1). The D1 ring of the heterohexamer is inactive[35,38,63] and ATP hydrolysis is limited in the D2 ring. Pex1 and Pex6 are arranged in a parallel manner to form oblique dimers that are capped by a tight Pex1(N2)/Pex6(N2) dimer, stabilized by strong electrostatic interactions (Fig. 2d). Our structures suggest the Pex1/Pex6 dimer is thus the building block of the heterohexameric assembly, throughout all layers of the complex, including the D2 ring. ATP plays a crucial role in assembly of the hexamer. All D1 binding pockets are occupied (Fig. 2) and previous studies have shown that the complex does not assemble in absence of ATP[64].

The molecular mechanism of how Pex1 and Pex6 work together to pull the peroxisomal receptor out of the peroxisomal membrane for another round of import has remained unknown. Our data substantially extend our knowledge about the molecular architecture and function of the yeast Pex1/Pex6 complex.

We used a Pex6 Walker B mutant (E832Q) and "trapped" the heterohexamer processing an endogenous substrate along the central channel formed by the active D2 ring. Our cryo-EM structures establish that the Pex1/Pex6 heterohexamer is at first glance a "canonical" unfoldase engaging the substrate via a spiral staircase of conserved pore loops 1 along the central pore of the D2 ring[44–47,50,57]. Similar to other ATPases, Pex1/Pex6 operates in processive manner, with conserved pore loops 1 engaging the substrate and pulling it towards the exit of the D2 channel[44–47,50,57]. The positions of the NTDs are also similar to the previous EM structures of Pex1/Pex6 determined in the

presence of ADP, ATPγS, several nucleotide analogues, and without protein substrate[35,37,38]. This suggests that the overall conformation of the unique tandem NTDs of Pex6 and the N2 domain of Pex1 is independent of nucleotide or substrate binding. In contrast, in p97, the characteristic conformational change of the N-terminal domains from a similar "down" to the "up" conformation is triggered by binding to ATP (D1 domains) and/or binding to substrate and N-terminal cofactors[65–68]. The flexible Pex1(N1) domain is not resolved in our cryo-EM structures and might be involved in interactions with the substrate and/or yet unknown co-factor or adaptor proteins[69,70]. In summary, t the N-terminal domains Pex1(N2), Pex6(N1), and Pex6(N2) most likely do not rearrange during the ATP hydrolysis cycle, as is the case with p97[65,66,71].

We were able to localize the substrate in the D2 channel but due to limited resolution, we were not able to reveal its identity. Furthermore, MS analysis did not allow us to unambiguously identify the substrate. We hypothesize that the observed density could be a mixture of different endogenous peptides, or the complex itself may undergo self-processing. The Pex1/Pex6 complex has been previously proposed to solely target ubiquitinated Pex5, but more recently it has been shown that the complex is also capable to process its membrane anchor Pex15[24], which binds to the N2 domain of Pex6[24,72] or the pexophagy inducing receptor Atg36[33]. These substrates were however not present in our preparation. We did not observe any substrate density in the D1 pore, suggesting that the substrate and the D1 domains might not establish stable contacts. Interestingly, for Cdc48/p97, poly-ubiquitinated Ub-Eos (Ub$_n$-Eos) was observed both in the D1 and D2 rings[44,47]. In contrast, the smaller substrate hexa-ubiquitin (Ub6) was resolved only in the D2 ring[44], similarly to our cryo-EM structures.

The fact that the hexamer is built of alternating Pex1/Pex6 heterodimers has a major impact on the mechanism of substrate processing. Our study revealed two distinct functional states of substrate engaged Pex1/Pex6. In the better resolved state (dubbed "single seam"), the inactive D1 ring is symmetric (Fig. 2) and resembles a trimer of Pex6(D1)/Pex1(D1) dimers. The catalytically active D2 shows a spiral arrangement that drives translocation (Fig. 3b, e), as previously described for other unfoldases[44–47,50,57]. Aromatic residues of five alternating Pex6(D2)- and Pex1(D2)- pore loops 1 arranged in a spiral staircase engage the substrate (every two amino acids of the substrate) (Fig. 3b). The sixth loop is dislocated and thus, does not contact the substrate. This loop is positioned between the lowest and the highest position of the spiral staircase ("seam"). Interestingly, Pex1(D2) and Pex6(D2) provide distinct pore loops 1 with different properties regarding substrate-grip and inter-pore-loop 1 coordination (Fig. 3b).

The second "twin-seam" state reveals two "seam" D2 domains forming a characteristic Pex6(D2)/Pex1(D2) pair that is detached from the spiral (Fig. 4c). The associated Pex6(D1)/Pex1(D1) dimer is tilted outward and the D1 channel is widened (Fig. 4a; Supplementary Fig. 7e, f). Intriguingly, the pore loops 1 of the "twin-seam" in the D2 ring are close together and almost at identical heights, which, at least to our knowledge, has not been observed for canonical ATPases bound to substrate (Fig. 4c). Our data clearly suggest that Pex1(D2) and Pex6(D2) function in concert with each other.

We propose a processive threading and inter-ring communication mechanism, shown schematically in Fig. 5, which involves Pex1(D2)/Pex6(D2) pairs and allosteric force propagation to the D1 ring. In the first step of the cycle, two Pex6(D2)/Pex1(D2) dimers are bound to the substrate (Fig. 5b, step i). The characteristic D2 "twin-seam" dimer is dislocated from the central channel of the complex. The associated Pex6(D1)/Pex1(D1) dimer on top of the D2 "seam-dimer" is aligned to this arrangement and is also dislocated from the central D1 channel (Fig. 5a, step i). This results in an "widened" conformation of the complex (Supplementary Fig. 7e).

To explain the transition from the "twin seam" to the "single seam" state, we propose that once the Pex1(D2) "seam" (F in Fig. 5b, c, step i)

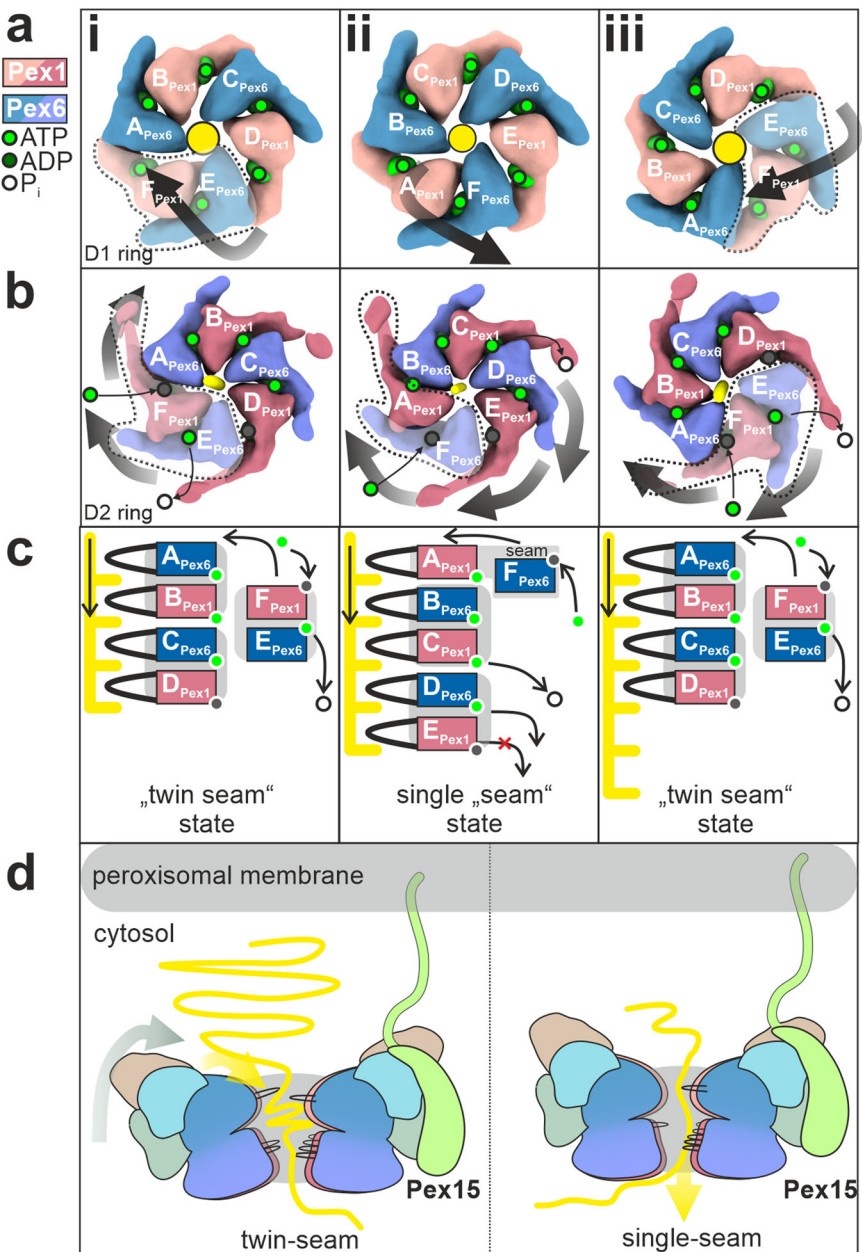

**Fig. 5 | Proposed model of Pex1/Pex6 assembly and substrate translocation.**
**a**–**c** Schematic of proposed Pex1/Pex6 model for ATP-fueled substrate processing. Top view of the D1 **a** and D2 ring **b** and schema of the pore1-loop arrangement **c** in "twin-seam"(i)→"single-seam"(ii)→"twin-seam"(iii) conformational transitions. The substrate is shown in yellow. The black arrow indicates the movements of the individual domains. The subunit occupying the highest position of the right-handed spiral staircase at the respective state, is defined as subunit "A". Pex1(D2) and Pex6(D2) are alternating in the D2 ring, thus subunit A can be either Pex1 or Pex6. Note that the ATPase does not rotate during the ATPase cycle, rather the ATP is hydrolyzed in a counterclockwise manner. Also note that both Pex6 and Pex1

form dimers both in D1 **a** and D2 **b** rings. Pex1(D2)/Pex6(D2) heterodimers are highly coordinated in processing the substrate. The D1 ring is inactive, but the Pex1(D1)/Pex6(D1) heterodimer tilts inwards in the "twin-seam"(i)→"single-seam"(ii) transition, whereas it swings out in"single-seam"(ii)→"twin-seam"(iii) transition. **d** The Pex1/Pex6 heterohexamer binds to Pex15 at the peroxisomal membrane, with the "mouth" of the D1 ring facing the peroxisome. In the "twin-seam" state, the D1 dimer opposing the spiral staircase (stacked on top of the twin seam), is tilted outward in an open conformation. Triggered by the power stroke in the D2 ring, the D1 dimer tilts inward and "pushes" the substrate towards the opening of the spiral staircase, supporting thereby substrate unfolding.

binds ATP and engages the substrate at the top of the spiral (moves to position A in Fig. 5b, c, step ii), the second domain of the D2 "seam" dimer, Pex6(D2)(E), follows towards the highest position of the staircase. However, it still does not contact the substrate and remains as a "single seam" subunit (becomes F in Fig. 5b, c, step ii).This is the step where the pair separates transiently during the ATP hydrolysis cycle. However, due to the "clicking" of the first pore 1 loop of the Pex1(D2) (A)/Pex1(D2)(F) dimer on the spiral, there are steric interactions at a "communication" hub between Pex6(D2) and Pex1(D1) within the

dimer (Fig. 4d). This triggers a swing-in of the Pex1(D1)/Pex6(D1) dimer stacked on the D2 "twin seam" and associated N-terminal domains toward the central channel, resulting in a "tightened" conformation of the D1 ring (Fig. 5a, step ii, Supplementary Fig. 7d). Pex1(D2)(E) (Fig. 5b,c, step ii) subsequently arrives at the bottom of the spiral. However, in the next step, this domain is not released from the spiral and does not become a "single seam" protomer, as would be the case in a canonical sequential ATPase cycle when hydrolysis at subunit D leads to the release of subunit E.

Instead, it remains tightly bound to its dimer partner Pex6(D2)(D) (Fig. 5b, c, step ii) because the nucleotide at position D (Pex6(D2)) is not hydrolyzed. Translocation of the substrate comes to a halt because the spiral cannot move (Fig. 5b, c, step ii). The ATPase skips this position and instead hydrolyzes the nucleotide at position C (Pex6(D2)), which is located between the dimers (Fig. 5b, c, step ii). The ATP hydrolysis at this position, provides thus a power stroke to release the Pex6(D2)(D)/ Pex1(D2)(E) dimer at the bottom of the spiral, which simultaneously allows the engagement of Pex6(D2)(F) (formerly "single seam") at the top of the spiral after it has bound to ATP (Fig.5b, c, step ii). The release of the "twin-seam" is propagated to the D1 ring and the associated D1 dimer swings out (Fig. 5a, step iii). Once the "twin seam" is released, hydrolysis is now possible at position E due to the loosening of the dimer interface (Fig. 5b, c, step iii). This in turn triggers the "twin seam"→"single seam" transition. In summary, our data cannot be reconciled with the traditional "hand-over-hand" model. The ATPase cycle of Pex1/Pex6 involves the release of "two hands" as the uncoupling of Pex6(D2) from the substrate is highly coordinated with Pex1(D2).

At first glance, the organization resembles that of Lon proteases, in which NTD dimers crown a homohexameric ATPase ring, resulting in a "trimer-of-dimers" arrangement[60,73,74]. However, the NTD dimers in Pex1/Pex6 are heterodimers formed by complementary charges between the domains of two different proteins (Fig. 2d, e). A characteristic "trimer of heterodimers" arrangement continues in the catalytically inactive heterohexamer in the D1 ring, which is absent in Lon proteases. While in Lon proteases steric conflicts lead to disentanglement of two seam subunits[60,73,74], in Pex1/Pex6 we observe disentanglement of a more compact "twin seam" dimer in the D2 ring with an ATP in the binding pocket between the seam subunits and the pore loops almost in the same plane (Fig. 4a, e). Pex1/Pex6 has thus developed unique structural features involving two distinct proteins organized in pairs in three layers to achieve its function, resulting in an adaptation of the otherwise conserved hand-over-hand mechanism.

To understand the underlying coordination mechanism of "twin-seam" release, we examined the conformation of the intersubunit signaling motif (ISS) in the "single-seam" state (Fig. 5c, step i). Recent studies on substrate-bound p97 revealed a conformation of ISS capable of controlling the nucleotide status of the adjacent subunit by projecting into the nucleotide binding pocket of the adjacent subunit and blocking ATP hydrolysis[44]. ATP hydrolysis occurs only once the ISS motif is retracted (Supplementary Fig. 13a)[44]. Thus, in p97, the ADP state of subunit E at the bottom of the spiral is communicated to the immediately adjacent subunit, triggering ISS-retraction and allowing the staircase to progress sequentially (Supplementary Fig. 13a)[44]. In Pex1/Pex6 "single-seam" state (Supplementary Fig. 13b), however, the D2 domains are organized as tight dimers and this signal is not passed on to the directly neighboring subunit, but two subunits further. On the one hand, the ISS motif of Pex1(E) remains inserted and prevents the hydrolysis of the directly neighboring Pex6(D). On the other hand, the ISS motif of Pex6(D) is in an intermediate conformation and is almost retracted from the nucleotide site of Pex1(C), which is located at the dimeric interface. Therefore, as expected, this site is predisposed to ATP hydrolysis, which explains the release of the "twin seam" Pex6(D)/Pex1(E) (Fig. 5c, step ii) instead of Pex1(E) as a "single seam." Most importantly, it is only after the release of the dimer (Fig. 5c, step iii) that the dimeric interface is loosened and the ISS motif of Pex1(F) is retracted, allowing hydrolysis at the dimeric interface (Supplementary Fig. 13c–e) (Pex6(E)) and continuation of the ATPase cycle. In summary, our data suggest that the ISS-motif is critical for the coordination of Pex1/Pex6 pairs and release of the "twin-seam".

However, an important question remains unanswered: "What is the functional significance of the opening of the D1 ring, powered by the D2 ATPase cycle?" A possible answer is that the "swinging" might be helpful to position and "push" the substrate towards the "mouth" of the central channel, thereby supporting the unfolding of the substrate (Fig. 5c). It should be emphasized that Pex1/Pex6 is so far lacking cofactors, known to facilitate correct orientation for substrate binding in other AAA-ATPases. Furthermore, the swing occurs in the D1 dimer, which opposes the spiral staircase in the underlying D2 ring (Supplementary Fig. 7f). In the absence of cofactors, inward movement and pore closure might thus be required to position the incoming substrate at the opening of the pore (Fig. 5d), exerting forces that support unfolding and making the ATPase more efficient.

The role of Pex1(N1) in this process remains unclear. This domain is flexible and not resolved in the cryo-EM density. It is expected to bind substrates or co-factors (similar to the N-domains of p97), but it is yet unclear if it binds ubiquitin[69]. It is also possible that the D1 tilting propagates forces to interaction partners. For example, it might also affect the membrane anchor of Pex15 via Pex6(N2) (Fig. 5d), which links the peroxisomal exportomer with the importomer and vice versa[15,28,75].

Recent cryo-EM data suggest that the peroxisomal Ub ligase complex, required for peroxisomal receptor recycling, might form a retro translocation channel at the peroxisomal membrane[20]. We expect that Pex1/Pex6 dimers assemble to hexamers near the peroxisomal membrane and bind via the Pex6(N2) domain to the elongated membrane anchor Pex15 (Fig. 5c)[24,43]. Pex15 would then orient the D1 ring facing towards the peroxisomal membrane, possibly in close proximity to the ligase complex. The ubiquitinated receptor Pex5 would then be taken up by the Pex1/Pex6 complex and pulled back into the cytosol[19] (Fig. 5c, yellow line). However, it is still unclear how the ubiquitinated Pex5 receptor is delivered to the ATPase channel and whether other cofactors and/or Pex1(N1) are involved.

In summary, this study reveals the architecture of the molecular motor of peroxisomal receptor recycling and suggests a mechanism of substrate translocation and utilizing communication between D1 and D2 rings, with fundamental differences to other type II AAA-ATPases. Our data establish a solid foundation for future studies towards understanding the complex interplay of the peroxisomal exportomer and importomer machinery.

## Methods
### Cloning
*S. cerevisiae* Pex6 (NCBI gene number 855387 containing the E832Q mutation) and Pex1 (NCBI gene number 853636, WT) were cloned with a Gibson assembly into a pESC-URA vector (Agilent technologies, Santa Clara, USA). An N-terminal His-tag and a C-terminal strep$_2$-tag have been cloned to Pex6(E832Q) and Pex1, respectively. All constructs were verified by sequencing of the inserts.

### Protein expression and purification
The generated plasmid was transformed via the lithium acetate method into MH 272/3fα *S. cerevisiae* cells[76] and successfully transformed cells were selected on SD media plates lacking uracil. One colony has been picked and inoculated in SD liquid media for preculture at 30 °C for 12 h. The preculture has been shifted to fresh SD media and incubated at 30 °C for 24 h. Co-expression was induced upon addition of 1/3 volume YEPG containing 2% (w/v) D-galactose. The expression has been further incubated at 30 °C for 16-17 h. Cells were sedimented at 5,000 rpm for 10 min. and 4 °C in an Avanti JC20XP (Beckman coulter, Brea, USA) centrifuge equipped with a JLA 8.100 (Beckman coulter, Brea, USA) rotor. The 35 g cell pellet was washed with deionized H$_2$O and either further processed or frozen at −80 °C.

For lysis, the cells were resuspended in lysis buffer (50 mM TRIS, pH 8, 300 mM NaCl, 5 mM MgCl$_2$, 1 mM ATP, 1 mM TCEP, 5% Glycerol, 1x Protease Inhibitor) and drop-wise frozen in liquid nitrogen to form small beads. The frozen cells were lysed in a ZM

200 ultra-centrifugal mill (Retsch GmbH, Haan, Germany) pre-cooled with liquid nitrogen at 18,500 rpm until a homogeneous powder was obtained.

The cell lysate was thawed, and debris were removed by centrifugation at 25,000 rpm (SA25.50 rotor, Beckman Coulter, Brea USA) for 45 minutes at 4 °C using an Optima XPN-80 (Beckman Coulter, USA). The supernatant was subsequently further cleared by an additional ultracentrifugation step at 40,000 rpm (SA25.50 rotor, Beckman Coulter, Brea, USA) for 15 minutes at 4 °C and filtration with a 0.45 μm nitrocellulose filter.

For strep-tactin purification, 300 ml filtered supernatant were subjected to 1.5 ml strep-tactin super flow high-capacity beads (IBA Lifesciences GmbH, Göttingen, Germany) in a gravity flow column. The beads were washed with 10 column volumes strep washing buffer (50 mM TRIS, pH 7.4, 300 mM NaCl, 3 mM ATP, 5 mM MgCl$_2$, 1 mM TCEP, 5% Glycerol). The protein was eluted with strep elution buffer (50 mM TRIS, pH 7.4, 300 mM NaCl, 1 mM ATP, 5 mM MgCl$_2$, 1 mM TCEP, 5% Glycerol, 10 mM desthiobiotin). Collected fractions containing the proteins of interest were concentrated using a 100 kDa MWCO centrifugal filter at 2,500 rpm and 4 °C.

The concentrated protein fractions were separated by size on a Superose 6 10/300 column operated with an AEKTA Purifier (GE Healthcare Bioscience, Chicago, USA). UV-absorption was monitored and Pex1/Pex6(E832Q) fractions were pooled and concentrated to a final protein concentration of 3.6 mg/ml.

To identify fractions of interest, Mini-PROTEAN TGX precast gels (Bio-Rad, USA) were used as recommended by the manufacturer. Protein bands were visualized with UV illumination or semi-dry blotted to Trans-Blot Turbo Transfer Packs (Bio-Rad, Hercules, USA). Immunolabeling was performed with horse-reddish peroxidase (HRP) coupled strep-tactin or anti-His antibodies.

### Single-pot solid-phase-enhanced sample preparation (SP3) for LC/MS

An SEC fraction containing co-purified Pex1-twinstrep and Hi-Pex6 (15 μg) was prepared using the Single-Pot Solid-Phase-enhanced Sample Preparation (SP3)[77]. After tryptic digestion and prior to MC/MS analysis samples were desalted on home-made 2 disc C18 StageTips as described[78]. After elution from the StageTips, samples were dried using a vacuum concentrator (Eppendorf) and the peptides were taken up in 10 μL 0.1% formic acid solution.

### LC-MS/MS

Experiments were performed on an Orbitrap Fusion Lumos (Thermo) that was coupled to an EASY-nLC 1200 liquid chromatography (LC) system (Thermo). The LC was operated in the one-column mode. The analytical column was a fused silica capillary (75 μm × 32 cm) with an integrated frit emitter (CoAnn Technologies) packed in-house with Kinetex C18-XB 1.7 μm core shell resin (Phenomenex). The analytical column was encased by a column oven (Sonation) and attached to a nanospray flex ion source (Thermo). The column oven temperature was adjusted to 50 °C during data acquisition. The LC was equipped with two mobile phases: solvent A (0.2% formic acid, FA, 2% Acetonitrile, ACN, 97.8% H2O) and solvent B (0.2% formic acid, FA, 80% Acetonitrile, ACN, 19.8% H2O). All solvents were of UPLC grade (Honeywell). Peptides were directly loaded onto the analytical column with a maximum flow rate that would not exceed the set pressure limit of 980 bar (usually around 0.6 – 1.0 μL/min). Peptides were subsequently separated on the analytical column by running a 60 min gradient of solvent A and solvent B (start with 3% B; gradient 3% to 9% B for 1:40 min; gradient 9% to 40% B for 37:40 min; gradient 40% to 100% B for 16:00 min and 100% B for 4:40 min) at a flow rate of 300 nl/min. The mass spectrometer was operated using Tune v3.3.2782.28. The mass spectrometer was set in the positive ion mode. Precursor

ion scanning was performed in the Orbitrap analyzer (FTMS; Fourier Transform Mass Spectrometry) in the scan range of m/z 300-1500 and at a resolution of 120000 with the internal lock mass option turned on (lock mass was 445.120025 m/z, polysiloxane)[79]. Product ion spectra were recorded in a data dependent fashion in the FTMS at resolution 7500 with auto scan range. The ionization potential (spray voltage) was set to 2.1 kV. Peptides were analyzed using a repeating cycle consisting of a full precursor ion scan (4.0 × 105 ions or 50 ms) followed by a variable number of product ion scans (5.0 × 104 ions or 22 ms) where peptides are isolated based on their intensity in the full survey scan (threshold of 25000 counts) for tandem mass spectrum (MS2) generation that permits peptide sequencing and identification. Cycle time between MS1 scans was 3 sec. Fragmentation was achieved by stepped Higher Energy Collision Dissociation (sHCD) (NCE 25, 30, 35). During MS2 data acquisition dynamic ion exclusion was set to 60 seconds and a repeat count of one. Ion injection time prediction, preview mode for the FTMS, monoisotopic precursor selection and charge state screening were enabled. Only charge states between +2 and +6 were considered for fragmentation.

### Peptide and Protein Identification using Proteome Discoverer 2.5

RAW spectra were submitted to an Sequest HT& MS Amanda 2.0 search in Proteome Discoverer (2.5.) using the default settings. The MS/MS spectra data were searched against the Uniprot one protein per gene (OPPG) S. cerevisiae reference database (UP000002311_559292_OPPG.fasta, 6060 entries, downloaded 12/1/2023). All searches included a contaminants database search (245 entries). The contaminants database contains known MS contaminants and was included to estimate the level of contamination. The searches allowed oxidation of methionine residues (16 Da) and acetylation of the protein N-terminus (42 Da) as dynamic modifications and the static modification of cysteine (57 Da, alkylation with cloroacetamide). Enzyme specificity was set to "Trypsin" with two missed cleavages allowed. The precursor mass tolerance was set to ±10 ppm and the MS/MS match tolerance was set to ±0.02 Da. The peptide spectrum match FDR and the protein FDR were set to 0.01 (based on target-decoy approach). Minimum peptide length was 6 amino acids.

### ATPase assay

For assaying ATP hydrolysis, 8.75 ng/μl of Pex1/Pex6$^{WB}$ purified from yeast was incubated with 100 μM ATP for 24 hours at 30 °C. Aliquots were collected at three time points: (a) immediately after ATP addition, (b) at 4.5 h, and (c) at 24 h of incubation. These samples were rapidly frozen in liquid nitrogen. Subsequently, the samples were boiled at 95 °C for 5 min and centrifuged at 16,900 ×g for 2 min. To quantify the amount of ADP produced, 6 μL from each sample was injected onto a ProntoSIL 120-5 C18 AQ column (Knauer) and subjected to a mobile phase consisting of 27 mM K2HPO4, 54 mM KH2PO4, 8 mM tetrabutylammonium bromide, and 20% acetonitrile, at a temperature of 40 °C and a flow rate of 0.8 ml/min. Ratios of ATP/ADP display the degree of ATP hydrolysis over time.

### Negative stain electron microscopy

Carbon coated copper grids (Agar scientific, Stansted Mountfitchet, United Kingdom; G2400C) were glow discharged as described previously[12] and then treated with Poly-L-lysine by incubating a droplet of 3 μl 0.1% (w/v) poly-L-lysine hydrobromide (Sigma-Aldrich, St. Louis, USA) for 30 seconds on the grid. Excess fluid was subsequently blotted with Whatman paper (No 5.), and the grids were washed two times with 10 μl water. After the grids were dried, 4 μl of purified Pex1-strep$_2$/His-Pex6(E832Q) complex was applied for 2 minutes at room temperature. The sample was blotted, washed

with 10 µl 0.75% uranylformate solution, and blotted again. Another 10 µl 0.75% uranylformate solution was applied to the grid and incubated for 45 seconds at RT.

Images were recorded with a JEM-1400 (JEOL, Akishima, Japan) equipped with LaB6 cathode and a 4 K CMOS detector F416 (TVIPS). Images were acquired at a pixel size of 1.84 Å.

## Cryo-EM sample preparation and data acquisition

For cryo-EM sample preparation, 4 µl protein solution (1.8 mg/ml) were applied to a glow-discharged holey carbon grid (R 2/1, 200 mesh, Quantifoil Micro Tools GmbH, Großlöbichau, GER). Grids were blotted for 3.5 sec and plunge-frozen after 1 sec drain time at 100% humidity in liquid ethane using a Vitrobot II (FEI, Hillsboro, USA). The plunged EM grids were used for automatically recording of 16,763 movies using the EPU software with a Titan Krios (FEI, Hillsboro, USA) with Cs-corrector and XEFG electron source at 300 kV.

## CryoEM data processing

The resolutions of cryo-EM maps are given according to the gold standard Fourier shell correlation (FSC) after combination of the half maps applying a full particle mask unless otherwise stated. During dataset acquisition, the quality of the data was monitored and the data was transferred using TranSPHIRE[80]. The super resolution movies (0.34 Å/pixel) were subjected to motion correction using Motioncor2 1.3.2 at 0.68 Å/pixel[81] and the Contrast Transfer Function (CTF) was estimated using CTFFIND 4.1.14[82]. Particles in representative micrographs were manually picked to train a model, which was then used for automated particle picking in crYOLO 1.8.1[83], selecting 1,259,079 particles with a box size of 384 ×384 pixels. Subsequently, 2D classification was performed using the iterative stable alignment clustering method (ISAC) with 200 particles/class[84,85]. After manually sorting out the "low quality" 2D classes, 605,359 particles remained. A 3D reconstruction (4.5 Å) was computed with MERIDIEN[86] using a negative stain EM structure (EMDB-2585) as reference[35]. The particles were further processed in RELION 3.1.4[87,88] using the reconstruction previously obtained with MERIDIEN. The particle overall demonstrated a pseudo C3-symmetry with more flexible N-terminal domains and an unsymmetric D2-ring. To improve overall resolution and avoid symmetry conflicts or particle misalignment during global 3D classification, particles were triplicated and given as an input for a 3D refinement imposing C3-symmetry reaching 4.5 Å. Subsequently, the C3-refined map was used as a template in 3D classification with C1-symmetry giving 6 classes. Then, duplicates have been removed from each individual 3D class. Finally, class 3 (147,925 particles) and class 4 (128,983 particles) representing two distinct conformations, were selected and further locally refined to an average resolution of 4.4 Å and 4.8 Å, respectively. Both classes were then improved with Bayesian polishing, per-particle CTF-refinement and another final round of Bayesian polishing to 4.1 Å (class 3) and 4.7 Å (class 4). The densities of class 3 and 4 were further improved with local anisotropic half map sharpening and density modification in Phenix 1.20[89,90] (Supplementary Table 1). The average resolution was improved to 3.9 Å (class 3) and 4.3 Å (class 4) resolution, respectively. Note that the resolution of density modified maps was determined via Phenix at FSC = 0.5[90]. The overall processing workflow is shown in Supplementary Fig. 3. Class 3 ("single seam") has an average resolution of 3.7 Å for the D1 ring and 3.9 Å (FSC = 0.143) for the D2 ring (improved to 3.3 Å (D1) and 3.6 Å (D2) after density modification) (Supplementary Fig. 4).

Class 4 shows an average resolution of 3.9 Å in the D1 and 4.7 Å in the D2 ring (improved to 3.8 (D1) to 4.3 (D2) Å after density modification) (Supplementary Fig. 12). Local resolution for all maps was estimated using Cryosparc 4. 3D FSC was computed using the 3DFSC server[91].

## Model building of Pex1/Pex6

Molecular models of the Pex1/6 complex are not available in the PDB. Alphafold predictions[92] of the full-length *Saccharomyces cerevisiae* Pex1 and Pex6 subunits have been used for rigid body fitting into the best resolved cryo-EM map (class3) and the assignment of the N1, N2, D1 and D2 domains of Pex1 and Pex6.

For model building of class 3 ("single seam"), the Alphafold predictions of Pex1 and Pex6 were first split to individual domains (N1, N2, D1, D2). Note that Pex1(N1) was not resolved in both cryo-EM structures. The individual domains of Pex1 (N2,D1,D2) and Pex6 (N1,N2,D1,D2) were fit to the density-modified class 3 map using ChimeraX 1.4[93] and the model was built with Coot 0.9.8[94]. The less well-resolved N-terminal domains Pex1(N2) and Pex6(N1) (class 3: ~6-8 Å) were instead only flexibly fitted in the density using the iMODFit plugin in UCSF Chimera 1.15[95,96]. The better resolved D1 (class 3: 3.3 – 3.8 Å) and D2 (class 3: 3.4–3.8 Å, seam subunit: ~6 Å) domains were real space refined into the density modified maps using Phenix 1.20[89]. For the protrusion domain of Pex1 (helix α28) bound to the N2-terminal domain of Pex6, an Alphafold multimer model was predicted for this interface (Supplementary Fig. 10c) due to low local resolution (~ 8 Å). This model was rigid-body fitted into the density. To avoid over-interpretation of highly flexible low-resolution regions, side chains were removed, but we retained Cα-connectivity and amino acid assignments for better clarity.

We observed a characteristic density in the channel of D2 rings corresponding to an unidentified endogenous substrate. We modeled a polyalanine peptide with an overall length of 9 residues. The density of our substrate is not well enough resolved to model a directionality of the substrate. Since both directions result in same model to map cross correlation values, the substrate was modeled with N-C directionality as it is the case for other AAA ATPases such as Cdc48[45,47].

Densities at the putative nucleotide binding domains of both cryo-EM structures have been carefully analyzed. We fitted either ATP or ADP argued by the density and opening of the nucleotide pocket or position of R-fingers (Supplementary Fig. 6). Mg$^{2+}$ ions were placed and relaxed in such a way that they enter into the favorable coordination with the ATP as well as the threonine of the Walker A domain using the ChimeraX molecular dynamics plugin ISOLDE[97]. In a last step, sterically unfavorable conformations have been corrected using Coot 0.9.8 with iterative rounds of Phenix 1.20 real-space refinements in between[89]. The entire procedure was repeated to create an atomic model for the less well resolved class 4 cryo-EM density. Here we used the molecular model obtained for class 3 as a starting point. The cryo-EM data collection parameters and refinement statistics are listed in Supplementary Table 2.

## Additional software

Images and movies were created using ChimeraX 1.4[93]. Electron density maps are shown at (class3, single-seam state) 4.42 sigma (2.26 sigma for density-modified) and at (class4, twin-seam state) 4.72 sigma (2.11 sigma for density-modified), respectively, with electron density not surrounding the model (> 4.5 Å distance) removed. Figure 4b was prepared with the PyMol script mod-evectors with a cutoff value of 4.0 Å. The length of the arrows depicts the magnitude of the motion. Distance and surface area measurements have been done using ChimeraX 1.4. Sequence alignments have been performed with SnapGene 6.0.6. Interfaces have been analyzed with the PDBePISA server[98]. Secondary structures of Pex1 and Pex6 (Supplementary Fig. 5) have been computed with PDBsum[99].

## Reporting summary

Further information on research design is available in the Nature Portfolio Reporting Summary linked to this article.

## Data availability

The cryo-EM maps of "single-seam" and "twin-seam" state have been deposited to the Electron Microscopy Data Bank (EMDB) under the accession codes (class3, single-seam state) EMD-16372 and (class4, twin-seam state) EMD-16373. The cryo-EM dataset has been deposited to EMPIAR under accession codes EMPIAR-11671. The coordinates of the corresponding models have been deposited to the Protein Data Bank (PDB) under accession codes (class3, single-seam state) 8C0V and (class4, twin-seam state) 8C0W. Mass spectrometry data are available via ProteomeXchange with identifier PXD043907. Source data are provided with this paper. Other data are available from the corresponding author upon request.

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

## Acknowledgements

This project has been supported by FOR 1905 (PerTrans consortium) GA 2519/1-2 to C.G. and ER 178/7-2 to R.E. and SFB1430, Project-ID 424228829 from the German Research Foundation (DFG) to C.G.. The cryo-EM dataset was processed at the Palma II HPC (DFG INST 211/667-1) of the University of Münster. We acknowledge technical support by the HPC team of the WWU IT, especially Holger Angenent and Sebastian Potthoff. We are grateful to Prof. Dr. Stefan Raunser for support and providing us access to the cryo-EM infrastructure of the Max Planck Institute (MPI) for Molecular Physiology Dortmund, Germany. We thank Dr. O. Hofnagel and Dr. D. Prumbaum (MPI Dortmund) for assistance with cryo-EM dataset acquisition. We thank Athina Drakonaki (Gatsogiannis Group, Institute of Medical Physics and Biophysics and Center for Soft Nanoscience, University of Münster) for kindly providing wild type p97 to establish and validate the ATPase assay. We thank Dr. Alexander Belyy (Max-Planck Institute Dortmund, Germany) for kindly providing us *S. cerevisiae* MH272/3fα wild type strain. We thank Prof. Dr. Hemmo Meyer for kindly providing us an antibody against Ubiquitin. C.G. is a member of the Cells-in-Motion Interfaculty center (CiM) and a faculty member of CiM-IMPRS, a joint graduate program of the International Max Planck Research School (IMPRS) Molecular Biomedicine and the Graduate School of CiM at the University of Münster. M.R. is a member of CiM-IMPRS.

## Author contributions

C.G. designed and managed the project. M.K., M.R. E.D.G. cloned constructs; M.K. established the expression and purification protocol with contributions from R.E.; M.K., M.R., E.D.G. performed protein purifications, screened samples, and collected negative stain EM data with input from P.L.; M.K. and P.L. collected cryoEM data; M.R. processed and analyzed cryoEM data with contributions from C.G. and B.U.K.; M.R. built the atomic model with contributions from BU.K; M.R. and E.D.G. performed ATPase assays; F.K. performed the mass spectrometry experiments and analyzed the results; M.R. prepared figures with input from C.G. and contributions from all authors; M.R. and C.G. wrote the manuscript. All authors discussed the results and commented on the manuscript.

## Funding

## Competing interests

The authors declare no competing interests.
