## [Peer Review File · Nature Communications]

Structure of the peroxisomal Pex1/Pex6 ATPase complex bound to a substrateREVIEWER COMMENTS

Reviewer #1 (Remarks to the Author):

The manuscript by Rüttermann et al. reports two structures of the Pex1/Pex6 AAA+ ATPase bound to unknown substrate. Previous studies established that Pex1/Pex6 assemble as a trimer of dimers into hexameric double-ringed structures, which stands in contrast to the more typical arrangement of homohexamers formed by six copies of a single protein in other AAA+ enzymes. The structures presented in this manuscript reveal interesting similarities and differences compared to other AAA+ ATPases. The D1 motor is observed to adopt an inactive planar conformation, consistent with studies that demonstrate its inability to hydrolyze ATP. In contrast, substrate is bound to the D2 ring and displays a conserved mode of binding compared to other AAA+ complexes: subunits are arranged in a spiral staircase in which conserved pore loop motifs grip the unfolding substrate within the central pore.

The authors propose an interesting mechanistic difference between Pex1/Pex6 and other AAA+ ATPases as a result of the tight coupling between Pex6-Pex1 dimers. In traditional AAA+ complexes, the “hand-over-hand” model is such that the bottom-most subunit of the spiral (subunit E) disengages from the substrate and its weakened interface with the preceding subunit enables the transition of the subunit back to the top of the staircase. In contrast, when Pex1 is at the bottom of the spiral, its tight association with Pex6 prevents its disengagement from the complex; in this state, the authors propose that ATP hydrolysis occurs two subunits above, i.e. at subunit C, resulting in the detachment of the Pex1-Pex6 dimer (subunits D and E). The overall mechanism is therefore distinct from other AAA+ ATPases in that the “hand” unit is defined as the Pex1-Pex6 dimer, rather than a single subunit.

Overall, the work is of value because Pex1/Pex6 are physiologically important, but there are major concerns about the quality of the structural work.

Major comments:

1. PDB validation statistics are of poor quality. The Ramachandran outliers and extensive clashes need to be fixed before publication. Of significant concerns are the clashes between nucleotide and surrounding elements, which undermine the confidence of nucleotide modeling.
2. Nucleotide assignments are inferred from measured distances between residues in Walker A and the arginine finger from the neighboring subunit as well as buried surface areas between subunits (Supplemental Fig. 5). Can the authors rule out the possibility that ATP can bind to Walker A without the neighboring arginine finger?

3. The nucleotide exchange model for Pex6 is difficult to conceptualize (Fig. 5c). What is the rationale that Pex6 hydrolyzes ATP after it disengages from the substrate? Based on the authors' model, it seems that Pex6 doesn't need to hydrolyze ATP at all.

4. ATPase assays should be shown to convey the extent of ATPase activity in complexes formed by the Pex6 D2 Walker B mutant. The manuscript cites relevant papers that report others' results, but it would be important to re-affirm whether ATPase activity is also completely abolished in their purified complex. On the same note, the Discussion describes D2 as being "catalytically active" (Page 23, Line 3), but the reported structures are assumed to be inactive.

5. Related to the above comment, if ATPase activity is completely abolished in the Pex6 D2 Walker B mutant, then what is the rationale for Pex1 being modeled as bound to ADP in both single- and twin-seamed states?

6. The substrate models for both states appear to extend beyond the observed densities, especially for the twin-seamed state. In particular, substrate density for the twin-seamed state appears very weak and discontinuous.

7. The substrate model for the twin-seamed state shows a strange loop at the bottom of D2 that is unexpected for a beta-strand conformation and is probably modeled incorrectly.

8. The significance of the twin-seamed state is overstated (Page 23, Line 16). The structure of the 19S proteasome also shows that two subunits can be simultaneously disengaged from the substrate (Zhang et al., Nature 2022), which seems conceptually similar to the twin-seamed state. The simplest explanation for these observations is that they are both intermediates of the hand-over-hand mechanism.

9. SDS-PAGE of the isolated complex indicate other prominent proteins. Can mass spectrometry proteomics be used to establish their identities. It would be interesting to confirm whether Pex5, Pex15, and/or Atg36 are present as potential substrates in authors' preps.

Minor comments:

1. The settings used for Phenix density modification need to be included, e.g. whether a model was used.
2. Relative threshold levels need to be reported for densities shown in Supplemental Figs 5a, b.
3. The local resolution heat maps shown in Supplementary Figs 3 & 10 are too hard to interpret. The color gradation between 4.5-8 Å needs to be made more clear, as well as labeling the subunits to indicate the components in D1/D2 that have variable resolutions. It would be particularly important to clarify the local resolutions of subunits E and F in D2.
4. Fig. 5 is difficult to understand. What are the thick black, red, and blue arrows intended to convey? What are the floating red bits in panel b?
5. FSC curves of maps before and after density modification need to be included as supplementary data.
6. Long segments/loops without supporting density should probably be removed from the model.

Reviewer #2 (Remarks to the Author):

Rüttermann et al provide a manuscript where they study the mechanism of Pex1/Pex6 substrate processing with cryo-EM. The manuscript is largely complete and comprehensive. It provides new insights into AAA+ ATPase processing, being the first high-resolution insights into how a heteromeric AAA processes substrates, in addition to peroxisomal biology. The team's findings will be of general interest. I have the following comments and suggestions below:

1. Page 6, Lines 10-12. Can the authors please clarify how the ATPase activity is compromised? Is it completely ablated? Reduced?

a. This could help clarify whether the trapped states are some sort of initiated state or a later step in processing.

2. If the authors used saturating ATP throughout the purification, why would the substrate still be engaged and not fully translocated?

a. The authors should test the ATPase activity of their purified complex.

b. Is it possible that the authors trapped some sort of initiation state, similar to the Cdc48 E588Q complex (Twomey, et al. 2018)?

3. Page 7, Line 15-16. How surprising is it that substrate was pulled down in sample preparation when expressing yeast proteins in yeast? The more surprising point is in point #2 above.

4. Cryo-EM maps in all figures: authors need to share what threshold the maps are displayed in each panel in the figure legends.

5. What are the functional consequences of the N2-N2 interactions? Does this have any effect on ATPase activity?

6. Nucleotide states: Can the authors clarify how nucleotides were assigned?

7. Supplementary Figure 5: The map sigma or threshold for each panel needs to be assigned, and the nucleotide density cannot be cut away or isolated from the rest of the map. The entire map should be shown at each position to give readers a realistic view into the nucleotide binding pockets. Threshold levels should be specified overall or for each panel if different because of local resolution.

8. Substrate directionality (page 14, line 5-7): can the authors please clarify what direction the substrate is going in, and label in Figure 3 panel b & e? Given the resolution, the authors should also provide a supplementary figure showing N-C and C-N fitting of the poly-A chain in the substrate density to provide rationale for the fit.

9. The authors need to show the cryo-EM density for all pore loops engaged around the substrate with the substrate density. This can either be integrated into Figure 3 or shown in a supplementary figure.

10. Supplementary Fig 1, a: what are the other major bands in the gel lanes? Can these be substrate? Possible to mass-spec those bands to ID. Have the authors tried western blot (i.e., against Ub?) to give some suggestion of the substrate identity?

11. I applaud the authors for deposition in EMPIAR.

12. Can the authors conceptualize their findings with the recent import paper (Gao...Rapoport, 2022) to give an overall view of peroxisome transport?

Reviewer #3 (Remarks to the Author):

Rutterman et al present cryo-EM structures of the Pex1/Pex6 AAA+ complex bound to substrate, providing insight into the molecular mechanisms underlying its translocase activity. Overall, the substrate-bound ATPases and the core mechanistic implications of these structures are essentially identical to those previously described for a wide variety of AAA+ proteins. However, given the distinct domain organization of the Pex1/Pex6 heterohexamer, these findings constitute an interesting addition to the extensive body of evidence supporting highly conserved mechanisms for AAA+ proteins. Moreover, these results shed light into the molecular principles underlying Pex1/Pex6 specific functionality, particularly regarding the organization of its unique N domains and the intersubunit communication interactions within the Pex1/Pex6 heterodimers. Still, this study would require major revisions prior to publication in Nature Communications.

Major points:

1. The novelty of this study stems from the identification of unique features of Pex1/Pex6, particularly the electrostatic network between the N2 domains and the communication hub around the nucleotide binding pocket. Can the authors confirm the functional relevance of their structural findings? For example, mutate some of the key residues involved in these interactions and biochemically characterize the mutants (e.g. complex assembly, ATP hydrolysis or translocase activity)?

2. Is it possible that the twin seam state is an artifact of the hybrid mutant complex (Walker B (WB) Pex6 but wild type Pex1) used for structural analysis? WB mutations delay hydrolysis. In FigS. 5b, the E subunit (WB Pex6) remains bound to ATP (has not yet hydrolyzed), whereas the D subunit (wtPex1) is bound to ADP (already hydrolyzed). This apparent “skip” of the sequential hydrolysis cycle might be a consequence of delayed hydrolysis of WB Pex6, and very unlikely in the context of the wild type complex. Did the authors study complexes with both Pex1 and Pex6 carrying WB mutations? Or complexes where both were wild type? Did they observe the double seam state in either case?

3. The resolution of the reconstructions and the quality of the derived atomic models are limited. While the resolution appears to be sufficient to support the authors' claims, further processing might help improve the resolution, which seems to be limited by unresolved sample heterogeneity and rigid body rotations of different parts of the complex relative to one another. In SFig. 2, it appears that the authors performed symmetry expansion. Did they ever do focused classification using a mask for a single heterodimer on the expanded dataset? Also, did they try masking individual rings or domains and doing focused classification on those? Or doing multibody refinement? Have the authors tried clustering the particles (3D Classification without alignment)? Or 3d variability analysis in cryosparc?

4. The authors describe the complex as a "trimer-of-dimers" and seem to conclude that this is due to the heterodimeric nature of the Pex1/Pex6 complex. However, a similar organization was also observed in Lon AAA+ protease, a homohexameric protein (Shin et al Nat. Comm 2021). Like in Pex1/Pex6, the N-domains of Lon appear to form a trimer of dimers and the substrate-bound Lon ATPases were found in a 4 ATP, 2 ADP state, despite being identical. Is it possible that N domain driven interactions might give rise to trimer-of-dimers organizations across different AAA+ proteins independent of homo- or heterohexamer?

5. I would recommend that the authors better separate their structural observations (results) from their conclusions (discussion). Throughout the manuscript, they propose some great interpretations, but I would encourage them to remove them from the results section and integrate them into an extended discussion regarding the implications of the unique structural features they identify in Pex1/Pex6. For example, the authors speculate in the results section that the position of the N domains might be nucleotide independent and cite previous studies. This would make sense to me, because the N2 domains sit atop the planar hydrolysis-incompetent D1 ring. However, I don't think the authors have enough evidence to support this statement in the results section, because they do not have any apo (nucleotide-free) subunits in their substrate-bound structures. Could the position of the N domains be the same in ADP and ATP states but different in an apo state? Can the D1 domains bind ADP or do they only ever bind ATP? Does a network of residues allosterically connect the N-domains to the nucleotide binding pocket (either D1 or D2) as seen in many other AAA+ proteins?

Minor points:

6. The authors mention interesting differences in ATP hydrolysis rates of the Pex1 vs Pex6 WB mutants. Could they add those results as a panel in a supplementary figure? Were the results consistent across diverse ATP concentrations?

7. The authors picked less than 2 million particles out of a huge dataset of 16000 micrographs. In the raw micrograph shown in FigS.1, many particles appear to be somewhat "disintegrated". Can the

authors comment on this? Did they observe different sample behavior when using different grids or freezing conditions?

8. I agree with the authors' conclusion that the single and double seam states are substrate-bound states within the substrate translocation cycle. To prevent confusion, I would recommend not to call it an "open" state in the discussion, as this is often used to refer to the substrate disengaged state (sometimes also called "off" state) with an open seam observed for other AAA+ proteins in the absence of substrate (e.g Hsp104 (Gates et al Science 2017) and Lon).

Reviewer #4 (Remarks to the Author):

Heterohexameric AAA+ ATPase Pex1/Pex6 complex is important for peroxisome biogenesis. Both of Pex1 and Pex6 are composed of tandem N1 and N2 domain followed by two ATPase domains (D1 and D2 domain). The D1 and D2 domains within Pex1/Pex6 can form two stacked hexameric rings, but only D2 domains are able to hydrolyze ATP. In this manuscript Rüttermann et al introduced Walker B mutation within the D2 domain of yeast Pex6 which reduced ATP hydrolysis while preserved the substrate binding, in beneficial of capturing translocation process. They further determined the cryo-EM structure of Pex1/Pex6 complex with an endogenous substrate bound in the D2 ring. From the cryo-EM data analysis, they analyzed two distinct conformations, ie single seam state Class 3 and twin seam state Class 4, and proposed a mechanism about how the Pex1/Pex6 functions to translocate the substrates in a right-handed spiral staircase manner via pore-1 loops. Overall, the manuscript appears to be appropriate interpretation on the structures with extensive and detailed discussion.

Specific comments:

- 1.The author focused on characterizing the better resolved Class 3 and Class4. However, there are 6 classes in total from 3D classification. Have the authors compared the details of the other 4 classes with Class 3 and Class4. Is it possible to differentiate more intermediate steps?
- 2.The structures in this paper suggested that ATP within D1 domains plays a role in stabilization the Pex1 D1 small ATPase domain rather than inter-molecular association. Additional biochemical assay would strengthen the claims.
- 3.The author showed that Pex1 (N2) and Pex2 (N2) can stabilize with each other via complementary electrostatic interaction. Could mutations in this interface abolished the assembly or activity of the complex?
- 4.Please define the meaning of saturating concentration of ATP in page 6?

5. In Supplementary Figure 1, the SDS-PAGE results showed besides of Pex1/Pex6, there are other obvious protein bands, what are the identity of these proteins? Are they potential substrates that were trapped in the pore?

6. The protein expression and purification section in Supplementary part, there are two tandem paragraphs described pooling the elutes and running gel filtration, one of them may be redundant.

7. The authors should include B factors for protein and ligand in Supplementary Table 1.

We thank the reviewers for their helpful comments on our work and useful guidance in revising our paper. Below is a point-by-point response to all comments and a detailed description of all changes we have made to our manuscript after considering their suggestions.

Reviewer #1 (Remarks to the Author):

The manuscript by Rüttermann et al. reports two structures of the Pex1/Pex6 AAA+ ATPase bound to unknown substrate. Previous studies established that Pex1/Pex6 assemble as a trimer of dimers into hexameric double-ringed structures, which stands in contrast to the more typical arrangement of homo-hexamers formed by six copies of a single protein in other AAA+ enzymes. The structures presented in this manuscript reveal interesting similarities and differences compared to other AAA+ ATPases. The D1 motor is observed to adopt an inactive planar conformation, consistent with studies that demonstrate its inability to hydrolyze ATP. In contrast, substrate is bound to the D2 ring and displays a conserved mode of binding compared to other AAA+ complexes: subunits are arranged in a spiral staircase in which conserved pore loop motifs grip the unfolding substrate within the central pore.

The authors propose an interesting mechanistic difference between Pex1/Pex6 and other AAA+ ATPases as a result of the tight coupling between Pex6-Pex1 dimers. In traditional AAA+ complexes, the “hand-over-hand” model is such that the bottom-most subunit of the spiral (subunit E) disengages from the substrate and its weakened interface with the preceding subunit enables the transition of the subunit back to the top of the staircase. In contrast, when Pex1 is at the bottom of the spiral, its tight association with Pex6 prevents its disengagement from the complex; in this state, the authors propose that ATP hydrolysis occurs two subunits above, i.e. at subunit C, resulting in the detachment of the Pex1-Pex6 dimer (subunits D and E). The overall mechanism is therefore distinct from other AAA+ ATPases in that the “hand” unit is defined as the Pex1-Pex6 dimer, rather than a single subunit.

Overall, the work is of value because Pex1/Pex6 are physiologically important, but there are major concerns about the quality of the structural work.

We thank the referee for the positive comments on our study and for insightful questions.

Major comments:

1. PDB validation statistics are of poor quality. The Ramachandran outliers and extensive clashes need to be fixed before publication. Of significant concerns are the clashes between nucleotide and surrounding elements, which undermine the confidence of nucleotide modeling.

We thank the reviewer for this comment. There are gradients of resolution in the density maps, and apparently we have not focused enough on improving the models in lower resolution regions/domains (e.g., the N-terminal domains). We apologize for this and we have now improved the models, which is reflected in the significantly improved statistics.

Before:

Model refinement		
	Class 3 Single-'seam' conformation	Class 4 Twin-'seam' conformation
Peptide chains	7	7
Residues	5566	5565
Ligands	ATP: 10, ADP: 2, Mg ²⁺ : 10	ATP: 10, ADP: 2, Mg ²⁺ : 10
RMSD Bond length (Å)	0.004	0.003
RMSD Bond angles (°)	0.745	0.733
Ramachandran outliers (%)	0.05	0.16
Ramachandran allowed (%)	9.47	13.49
Ramachandran favored (%)	90.47	86.34
Rotamer outliers (%)	0.04	0.02
MolProbity score	2.15	2.31
Clash score	12.53	14.9
EMRinger score	1.53	0.95
B-factors mean (Protein)	155.34	172.00
B-factors mean (Ligand)	93.89	127.56

Now:

Model refinement		
	Class 3 Single-'seam' conformation	Class 4 Twin-'seam' conformation
Peptide chains	7	7
Residues	5564	5564
Ligands	ATP: 10, ADP: 2, Mg ²⁺ : 10	ATP: 10, ADP: 2, Mg ²⁺ : 10
RMSD Bond length (Å)	0.003	0.004
RMSD Bond angles (°)	0.751	0.881
Ramachandran outliers (%)	0.04	0.00
Ramachandran allowed (%)	7.26	9.8
Ramachandran favored (%)	92.68	90.20
Rotamer outliers (%)	0.06	0.00
MolProbity score	1.81	1.89
Clash score	6.27	6.28
EMRinger score	1.53	0.99
B-factors mean (Protein)	132.26	131.73
B-factors mean (Ligand)	92.38	89.13

Clashes in the nucleotide binding pockets mainly involved the Mg²⁺. These ions are not resolved, but we still placed them approximately in the expected position for clarity. This has now been done even more carefully. We carefully checked the nucleotide modeling and could not find any other problems.

We updated the respective supplementary table 2 in the manuscript.

2. Nucleotide assignments are inferred from measured distances between residues in Walker A and the arginine finger from the neighboring subunit as well as buried surface areas between subunits

(Supplemental Fig. 5). Can the authors rule out the possibility that ATP can bind to Walker A without the neighboring arginine finger?

For the higher resolution 'single seam' state, we are confident in our ADP assignment based on density and confirmed by measured distances (see Supplementary Figure 6 d,e). Regarding the lower resolution 'twin seam' condition, we agree with the reviewer that based on our structure alone, we cannot rule out the possibility of ATP binding to Walker A without the adjacent arginine finger for the nucleotide pockets interpreted as ADP, particularly pocket D2 of subunit E. However, we would like to emphasize the following points. Mutations of either one of the arginine fingers (Pex1 or Pex6) results in a complex assembly defect as demonstrated by Ciniawsky *et al.* (2015) when using ATP as the nucleotide. Furthermore, the large number of available ATPase cryoEM structures with ADP in the nucleotide binding pockets of the ATPase ring, including 7SWL, 7T0V, 6AZ0, 7UQI, 7PX9, 7UPT, 6W22, 6W23, 6W24, 6P07, 7ABR, 7TDO and 6SH3, exhibit a consensus regarding the extent of opening of the nucleotide-binding pocket and the position of the loop of the respective 'ADP'-bound subunits in the spiral staircase (either at the bottom of the spiral or disengaged). In all available cryoEM structures, ATP, after binding to Walker A, also binds to the arginine finger and stabilizes the pocket, resulting often in better resolution compared to ADP. Consequently, the degree of opening of the nucleotide binding pocket is considered a reliable and well-established indicator of the nucleotide status of the respective subunit. We hope the reviewer agrees: it is extremely unlikely that ATP is bound to the Walker A motif of a wide-open (conserved) pocket without the neighboring arginine finger in a subunit that is disengaged from the spiral and detached from the substrate.

We rephrased this paragraph in the manuscript to make this point more clear:

The four D2 domains at the top of the spiral are ATP-bound (Pex1(A), Pex6(B), Pex1(C), Pex6(D) (Figure 3b,d; Supplementary Figure 6a; lower insets). Estimation of the nucleotide in the binding pocket of the 'bottom' Pex1(D) and 'seam' subunit Pex1(F) (Figure 3b,d) based on cryo-EM density alone was not possible with absolute confidence (Supplementary Fig. 6a). The large number of available AAA-ATPase cryoEM structures with ADP bound to one or more binding pockets (PDB: 7SWL, 7T0V, 6AZ0, 7UQI, 7PX9, 7UPT, 6W22, 6W23, 6W24, 6P07, 7ABR, 7TDO, and 6SH3) show however a consensus regarding the position of the respective pore loops 1 in the spiral staircase (bottom or disengaged) and the opening of the nucleotide binding pocket. The extent of nucleotide binding pocket opening has been thus established as a reliable indicator of the nucleotide status of the respective subunit^{45,49,62}. We therefore measured the buried surface area between adjacent protomers and distances between Walker A and arginine-fingers of the associated protomers (Supplementary Fig. 6c-e). In line with these observations, our analysis suggests that both the "bottom" Pex1(D) and the disengaged "seam" Pex1(F) D2 domains are indeed in an ADP-bound state (Fig. 3c,e, Supplementary Fig. 6a,c-e)^{43,46,63}.

3. The nucleotide exchange model for Pex6 is difficult to conceptualize (Fig. 5c). What is the rationale that Pex6 hydrolyzes ATP after it disengages from the substrate? Based on the authors' model, it seems that Pex6 doesn't need to hydrolyze ATP at all.

According to our model, ATP hydrolysis by Pex6 is not required for the disengagement but rather for the "re-engagement" of the "twin-seam" to the top of the spiral (see Fig. 5c, middle and right panels). When ATP is still bound at the interface between Pex1 and Pex6 in the disengaged Pex1/Pex6 dimer, this interface is "tight," and the dimer assumes a "compact" conformation that prevents the binding of Pex1 to the spiral and the continuation of the cycle.

4. ATPase assays should be shown to convey the extent of ATPase activity in complexes formed by the Pex6 D2 Walker B mutant. The manuscript cites relevant papers that report others' results, but it would

be important to re-affirm whether ATPase activity is also completely abolished in their purified complex. On the same note, the Discussion describes D2 as being “catalytically active” (Page 23, Line 3) but the reported structures are assumed to be inactive.

We appreciate the reviewer's reasonable request. Previous studies (Ciniawsky et al. 2015, Gardner et al. 2015) have reported that the WT/Pex6_WB complex abolishes ATPase activity, while Blok et al. (2015) demonstrated residual basal ATP hydrolysis. However, these studies employed indirect methods such as phosphate release (Ciniawsky and Blok) or ATP recovery (Gardner) over short time periods, and they examined complexes expressed in bacteria. To address the reviewer's comment, we conducted an HPLC-based assay to qualitatively assess ATPase activity by integrating peaks corresponding to ATP and ADP. Specifically, we incubated 350 ng of purified WT/Pex6_WB from yeast with 100 μ M ATP for 24 hours at 30°C. Aliquots were collected at three time points: a) immediately after ATP addition, b) at 4.5 hours, and c) at 24 hours of incubation. These samples were rapidly frozen in liquid nitrogen. Subsequently, the samples were boiled at 95°C for 5 minutes and centrifuged at 16,900 xg for 2 minutes. To quantify the amount of ADP produced, 6 μ L from each sample was injected onto a ProntoSIL 120-5 C18 AQ column (Knauer) and subjected to a mobile phase consisting of 27 mM K₂HPO₄, 54 mM KH₂PO₄, 8 mM tetrabutylammonium bromide, and 20% acetonitrile, at a temperature of 40°C and a flow rate of 0.8 ml/min. Regrettably, we encountered difficulties in performing the same experiment with WT Pex1/Pex6. Despite multiple attempts, we were unable to successfully express the WT complex in yeast, and we apologize for not being able to provide corresponding control measurements. Nonetheless, we have verified the assay's efficiency using wild-type p97, a highly active type II AAA ATPase, which is available in our laboratory. Our data suggest that the Pex1WT/Pex6_WB-variant complex indeed exhibits residual basal ATP hydrolysis activity, consistent with the findings of Blok et al. (2015). Thus, our results suggest that the ATPase activity is not completely abolished in this variant but significantly attenuated. However, this reduction in activity proved advantageous for "trapping" the AAA+ motor in a substrate-engaged state for structural determination, a strategy also employed for other ATPase complexes.

We added a new supplementary figure 1 in the manuscript and the description of the experiment in the Methods section. Furthermore, we added following sentence in the manuscript:

When Pex6 D2 carries a WB mutation, the complex maintains a residual basal ATPase activity (Supplementary Figure 1). The attenuated hydrolytic activity has proven to be advantageous in "trapping" the AAA+ motor in a substrate-engaged state for our structural studies similar to other AAA-ATPases^{48,48-50}.

Rebuttal Figure 1. Time-dependent decrease in ATP/ADP ratio analyzed by HPLC. (a-d) Representative chromatograms are shown. Control experiments were performed with WT p97(+ATP) and ATP alone. The ATP/ADP ratios in the presence of Pex1/Pex6_WB were obtained by quantification of the results from HPLC analyses at different timepoints upon addition of ATP to the protein sample.

5. Related to the above comment, if ATPase activity is completely abolished in the Pex6 D2 Walker B mutant, then what is the rationale for Pex1 being modeled as bound to ADP in both single- and twin-seamed states?

Please see our reply to comment 4. The ATPase activity is not completely abolished; we measure a clear ADP peak by HPLC after 24h upon addition of ATP to the Pex1/Pex6_WB complex.

6. The substrate models for both states appear to extend beyond the observed densities, especially for the twin-seamed state. In particular, substrate density for the twin-seamed state appears very weak and discontinuous.

We sincerely apologize for this and want to assure you that we have made significant improvements to the substrate models based on your comment. In the twin-seam state, the resolution is lower, as the substrate density interacts with only four loops and there is a wider pore opening. Therefore, considering in addition the overall lower resolution of the twin-seam state, it is not surprising that the substrate density appears less well resolved compared to the single-seam state. However, we have now included a new supplementary figure 11 showing the substrate model surrounded by its density at a given sigma value. Figure 3b and 4c have been updated.

7. The substrate model for the twin-seamed state shows a strange loop at the bottom of D2 that is unexpected for a beta-strand conformation and is probably modeled incorrectly.

We thank the reviewer for this comment. We corrected the model to the expected beta-strand conformation. Furthermore, we added a new supplementary figure 11 to show the substrate density and the pore loops.

8. The significance of the twin-seamed state is overstated (Page 23, Line 16). The structure of the 19S proteasome also shows that two subunits can be simultaneously disengaged from the substrate (Zhang et al., Nature 2022), which seems conceptually similar to the twin-seamed state. The simplest explanation for these observations is that they are both intermediates of the hand-over-hand mechanism.

We appreciate the reviewer's comment and would like to respectfully present our differing perspective. While it may initially appear that Zhang et al. depict a similar condition with two disengaged pore loops, there are significant distinctions. In Zhang et al.'s study, the pore loops are spaced approximately 1.5nm apart, and the released subunits do not form a dimer. Additionally, the interface between the subunits is relatively "loose," with the nucleotide binding pocket in an "apo" state.

In contrast, the "twin seam" state of Pex1/6 exhibits remarkable dissimilarities. The released subunits form an intriguingly compact dimer stabilized by ATP, and the respective pore loops lie almost in the same plane. These characteristic differences strongly suggest that a different conformation is described in the 19S proteasome, suggesting that the respective subunits are released gradually and independently, following the hand-over-hand mechanism. In Pex1/6, on the other hand, the respective subunits are clearly released as a dimer, as proposed in our model.

Furthermore, it is important to note that the twin seam conformation in our dataset is not an insignificant occurrence or a rare intermediate state but represents a substantial state, accounting for 43.9% of the total number of refined particles.

In our research, we utilized the Pex1/Pex6_WalkerB variant to effectively capture a substrate in the central pore. Importantly, the "twin seam" state, is not just a feature of the Pex6_WB variant. This state, characterized by an asymmetric D2 ring comprising three Pex1/Pex6 pairs, with one pair tilted outward (referred to as a twin seam dimer in our substrate-bound cryoEM structure), has also been consistently observed as the primary conformation in previous low-resolution cryoEM studies of **wild-type** substrate-free Pex1/6 (ATPyS) conducted by the Rapoport/Walz laboratories (see Blok et al., 2015, PNAS; Figure 1 and Rebuttal Figure 2). In addition, the twin seam state was also observed in negative stain reconstructions of WT Pex1/Pex6 (Gardner et al., 2015).

Furthermore, in response to Comment 2 of Reviewer 3, we collected a small cryoEM dataset of the Pex1_WB/Pex6_WB complex (WB mutation both in Pex1 and Pex6). 3D classification revealed the twin-seam state as the major conformation (51% of particles in the final particle set (Rebuttal Figure 4b).

Thus, this distinct pairwise arrangement and the tilting of a single Pex1/Pex6 D2 dimer away from the central pore constitute vital and unique features of Pex1/6, rather than merely representing an artifact of the Pex1/Pex6_WB variant nor an intermediate state within the typical hand-over-hand cycle observed in other type II ATPases.

Rebuttal Figure 2. Section through the D2 ring of Pex1/Pex6_WB (right image; this study) and WT_Pex1/Pex6 (EMDB 6359; left image; also shown in Figure 4, panel C, Blok et al., 2015, PNAS). Note the characteristic organization of WT Pex1/Pex6 (ATPyS) into three Pex1/Pex6 pairs, with one pair characteristically tilting outward similar to our twin-seam state (right). For direct comparison, we filtered our cryoEM structure to the nominal resolution of EMDB 6359 (~7 Å).

9. SDS-PAGE of the isolated complex indicate other prominent proteins. Can mass spectrometry proteomics be used to establish their identities. It would be interesting to confirm whether Pex5, Pex15, and/or Atg36 are present as potential substrates in authors' preps.

We performed multiple purifications of the Pex1/Pex6_WB complex from yeast and conducted mass spectrometry proteomics analysis on the samples. The results revealed several abundant hits, including Hsp60, Hsp26, Acetyl-CoA carboxylase, Ubiquitin-60S ribosomal protein, and Elongation factor 1-alpha. Unfortunately, our analysis was not able to unambiguously reveal the ID of the substrate. These proteins are known to be non-peroxisomal and there is currently no additional evidence for their physiological relevance as Pex1/Pex6 substrates. We want to avoid any over-interpretation, as (at least) some of these hits could be non-specific and/or contaminants.

Surprisingly, the expected candidates for Pex1/Pex6 substrates, such as Pex15, Atg36, and Pex5, were completely absent. It is important to note that the complex was overexpressed under galactose-induced conditions, which may lead to an imbalance in stoichiometry between peroxisomal proteins and the recombinant complex. A true substrate should ideally be present in a 3:1 stoichiometry with both Pex1 and Pex6.

Considering these findings, we hypothesize that the observed electron density could be a mixture of different peptides, or the complex itself may undergo self-processing. For instance, the flexible N-terminal domain of Pex1, which remains unresolved in our density, could play a role. In conclusion, at this stage, it is unfortunately not possible to definitively identify the substrate using mass spectrometry analysis.

We added following sentence in the main manuscript:

Qualitative mass spectrometry analysis of the recombinantly expressed Pex1/Pex6 complex did not allow us to unambiguously identify the substrate, as all peptides were non-peroxisomal and could therefore be non-specific substrates. Interestingly, known substrates such as Pex5, Pex15 of Atg36 were not present in our preparation. It is important to note that the complex was overexpressed under galactose-induced conditions, which may lead to an imbalance in the stoichiometry between peroxisomal proteins and the recombinant complex. A true substrate should ideally be present in a 3:1 stoichiometry with both Pex1 and Pex6. The density probably corresponds either to a mixture of several endogenous substrates trapped in the D2 pore during purification, or to an event of self-unfolding of the complex, as was recently observed in the cryoEM structures of Rix7⁴⁸ and VAT⁵².

Minor comments:

1. The settings used for Phenix density modification need to be included, e.g. whether a model was used.

The Phenix command prompt (v 1.20-4459) was utilized due to its enhanced feature set compared to the GUI. It is important to emphasize that no model was employed for density modification to prevent any potential bias introduced by a model. The specific prompts that were used are provided below and included now in the methods section as supplementary table 1:

	Class 3 Single-'seam' conformation	Class 4 Twin-'seam' conformation
Local anisotropic half-map sharpening	phenix.local_aniso_sharpen run_half1_class001_unfil.mrc run_half2_class001_unfil.mrc resolution=4.14	phenix.local_aniso_sharpen run_half1_class001_unfil.mrc run_half2_class001_unfil.mrc resolution=4.7 local_sharpen=True

	<pre> local_sharpen=True sharpened_map_file_1=.\\Output_command\\half1_local_sharp.ccp4 sharpened_map_file_2=.\\Output_command\\half2_local_sharp.ccp4 box_size_grid_units=384 sharpened_map_file=.\\Output_command\\full_local_sharp.ccp4 </pre>	<pre> sharpened_map_file_1=.\\Output_command\\half1_local_sharp.ccp4 sharpened_map_file_2=.\\Output_command\\half2_local_sharp.ccp4 box_size_grid_units=384 sharpened_map_file=.\\Output_command\\full_local_sharp.ccp4 </pre>
Density modification	<pre> phenix.resolve_cryo_em_half1_local_sharp_aniso.ccp4 phenix.resolve_cryo_em_half2_local_sharp_aniso.ccp4 seq_file=.\\Input\\Pex1_Pex6_yeast.fasta resolution=4.14 blur_by_resolution=True blur_by_resolution_factor=2 box_cushion=15 b_iso=90 dm_resolution=3.6 </pre>	<pre> phenix.resolve_cryo_em_half1_local_sharp.ccp4 phenix.resolve_cryo_em_half2_local_sharp.ccp4 seq_file=.\\Input\\Pex1_Pex6_yeast.fasta resolution=4.14 blur_by_resolution=True blur_by_resolution_factor=20 box_cushion=15 b_iso=90 dm_resolution=3.8 </pre>

2. Relative threshold levels need to be reported for densities shown in Supplemental Figs 5a, b.

Done. The map sigma for each figure was set to (class3) 4.42 (2.26 for density-modified) and (class4) 4.72 (2.11 for density modified), respectively and not changed between different panels. This is now stated in the materials and methods section.

3. The local resolution heat maps shown in Supplementary Figs 3 & 10 are too hard to interpret. The color gradation between 4.5-8 Å needs to be made more clear, as well as labeling the subunits to indicate the components in D1/D2 that have variable resolutions. It would be particularly important to clarify the local resolutions of subunits E and F in D2.

We thank the reviewer for pointing this out. We improved the quality of Supplementary Figures 4, 12 according to the suggestions of the reviewer.

4. Fig. 5 is difficult to understand. What are the thick black, red, and blue arrows intended to convey? What are the floating red bits in panel b?

Thick black arrows have been used to indicate the inward and outward tilting of a Pex1/Pex6 dimer within the D1 ring, depending on the state. Red and blue arrows were initially utilized to represent the tilting of Pex1 and Pex6, respectively, within the D2 ring. We acknowledge that this arrangement might be confusing. To address this concern, we have replaced the red and blue arrows with thick black arrows and provided a clear explanation of the arrows in the figure's legend.

The "plastic"-like surfaces of Pex1 and Pex6 were generated by filtering down the molecular models to a specific resolution (20 Å). Helix α 28 of Pex1, which connects the Pex6(N1) and Pex1(D2) domains (refer to Supplementary Figure 10c), is linked to the D2 domain of Pex1 through long, flexible loops that were not resolved in the model. Consequently, the simulated density appears discontinuous, and at first glance, α 28 may seem like a floating red bit. We acknowledge that this can be perplexing.

To mitigate any confusion, we have connected the densities in the illustration using red-transparent lines.

5. FSC curves of maps before and after density modification need to be included as supplementary data.

We added the FSC curves as suggested by the reviewer in Supplementary Figure 4 and 12. Please note, we will submit in the EMDB the maps both before and after modification.

6. Long segments/loops without supporting density should probably be removed from the model.

For better clarity, we decided against removing all loops that do not show any density. Based however on our alpha-fold predictions, we expect that the position and overall folding of these loops will be however approximately correct. Keeping those is in general helpful for having a better overall overview of the structure, but in parallel, we have to make sure that any over interpretations of our structures will be avoided. Therefore, to make clear that these loops are not resolved, we now kept the Ca atoms (and sequence context) but removed **all side chains**. This is now described in the Methods section.

Reviewer #2 (Remarks to the Author):

Rüttermann et al provide a manuscript where they study the mechanism of Pex1/Pex6 substrate processing with cryo-EM. The manuscript is largely complete and comprehensive. It provides new insights into AAA+ ATPase processing, being the first high-resolution insights into how a heteromeric AAA processes substrates, in addition to peroxisomal biology. The team's findings will be of general interest.

We thank the reviewer for the positive evaluation of our manuscript.

I have the following comments and suggestions below:

1. Page 6, Lines 10-12. Can the authors please clarify how the ATPase activity is compromised? Is it completely ablated? Reduced?

a. This could help clarify whether the trapped states are some sort of initiated state or a later step in processing.

We greatly appreciate the reviewer for raising this insightful question. Based on our new data (HPLC-based assay, as detailed in Reviewer 1, comment 4; Rebuttal Figure 1, new Supplementary Figure 1), we have observed that the ATPase activity of the Pex1 WT/Pex6 WB mutant complex, purified from yeast, is noticeably low but not completely abolished. There remains a residual basal ATPase activity present. From these findings, we rather hypothesize that we "capture" the complex at an early stage of substrate processing.

2. If the authors used saturating ATP throughout the purification, why would the substrate still be engaged and not fully translocated?

Most of the published structures of substrate-bound AAA+ translocases to date have utilized nucleotide analogs or Walker B mutants to hinder hydrolysis and effectively generate "substrate traps." This approach has proven invaluable in capturing the enzymes at intermediate stages of translocation. See for example:

S. N. Gates, A. L. Yokom, J. Lin, M. E. Jackrel, A. N. Rizo, N. M. Kendersky, C. E. Buell, E. A. Sweeny, K. L. Mack, E. Chuang, M. P. Torrente, M. Su, J. Shorter, D. R. Southworth, Ratchet-like polypeptide translocation mechanism of the AAA+ disaggregase Hsp104. *Science* 357, 273–279 (2017).

Cooney, I. et al. Structure of the Cdc48 segregase in the act of unfolding an authentic substrate. *Science* (New York, N.Y.) 365, 502–505; 10.1126/science.aax0486 (2019).

Puchades, C. et al. Structure of the mitochondrial inner membrane AAA+ protease YME1 gives insight into substrate processing. *Science* (New York, N.Y.) 358, eaao0464; 10.1126/science.aao0464 (2017).

N. Monroe, H. Han, P. S. Shen, W. I. Sundquist, C. P. Hill, Structural basis of protein translocation by the Vps4-Vta1 AAA ATPase. *eLife* 6, e24487 (2017).

Z. A. Ripstein, R. Huang, R. Augustyniak, L. E. Kay, J. L. Rubinstein, Structure of a AAA+ unfoldase in the process of unfolding substrate. *eLife* 6, e25754 (2017).

C. Deville, M. Carroni, K. B. Franke, M. Topf, B. Bukau, A. Mogk, H. R. Saibil, Structural pathway of regulated substrate transfer and threading through an Hsp100 disaggregase. *Sci. Adv.* 3, e1701726 (2017).

C. Alfieri, L. Chang, D. Barford, Mechanism for remodelling of the cell cycle checkpoint protein MAD2 by the ATPase TRIP13. *Nature* 559, 274–278 (2018).

Lo, Y.-H. et al. Cryo-EM structure of the essential ribosome assembly AAA-ATPase Rix7. *Nature communications* 10, 513; 10.1038/s41467-019-08373-0 (2019).

In our study of Pex1/Pex6, we successfully employed the same strategy. Therefore, it is expected that the substrate is not fully translocated, as the reduced hydrolysis rate significantly slows down the process. Furthermore, Gardner *et al.* (2018) demonstrated that this is indeed the case for Pex1/Pex6_WB using Pex15 as a substrate.

a. The authors should test the ATPase activity of their purified complex.

We thank the reviewer for this comment. We performed a HPLC-based assay to measure the time-dependent decrease of ATP/ADP in presence of Pex1/Pex6_WB. Please see Rebuttal Figure 1 and new Supplementary Figure 1 (detailed in Reviewer 1, comment 4).

b. Is it possible that the authors trapped some sort of initiation state, similar to the Cdc48 E588Q complex (Twomey, et al. 2018)?

As mentioned above, the Walker B mutation significantly slows down hydrolysis and effectively creates a "substrate trap". It is indeed unlikely that the available snapshots represent the full conformational landscape of active Pex1/Pex6 and due to attenuated hydrolysis rate, we might have indeed trapped an initiation state. Nevertheless, the presence of two major conformational states and the polypeptide in the central pore provides important new mechanistic insights into substrate threading and processing by Pex1/Pex6 and allows detailed comparison with structural studies on other AAA complexes employing similar strategies.

3. Page 7, Line 15-16. How surprising is it that substrate was pulled down in sample preparation when expressing yeast proteins in yeast? The more surprising point is in point #2 above.

This finding was still unexpected because we did not establish yeast growth conditions where the expected peroxisomal substrates (e.g. Pex5) might be in stoichiometry to the recombinant complex. Please see also Reviewer 1, Comment 9. We rephrased this sentence in the main manuscript.

4. Cryo-EM maps in all figures: authors need to share what threshold the maps are displayed in each panel in the figure legends.

Thank you for the suggestion. Sigma values have been included in the respective panels.

5. What are the functional consequences of the N2-N2 interactions? Does this have an effect on

Thank you for the question. Indeed, Ciniawsky et al. (2015) report that Pex6 N-terminal deletions result in unstable protein expression or compromised hexamerization.

6. Nucleotide states: Can the authors clarify how nucleotides were assigned?

We modeled the nucleotides according to the respective density in the cryoEM volume. ATP-bound subunits, typically showed for example clear density matching ATP and were also “closed” (as expected, because ATP binds the WalkerA, WalkerB and arginine finger of the adjacent subunit).

In case the density was not sufficiently resolved (for example fragmented density of subunit F or D D2 in twin seam state) we modeled according to the opening of the nucleotide binding pocket and position of the respective pore loop in the spiral staircase. We have to take in account, that in the meanwhile, there is a large number of substrate engaged AAA ATPase structures solved. The extent of nucleotide binding pocket opening has been thereby a reliable and well-established indicator of the nucleotide status. In addition, ADP-bound subunits display pore-loops that either positioned at the bottom of the spiral of disengaged from the substrate. In general, ADP does not bind to the arginine finger of the adjacent subunit and thus, the binding pocket is “open”.

In our density maps, pockets with fragmented nucleotide densities were also “open” and either at the bottom of the spiral or disengaged from the substrate. Therefore, they were modeled as ADP-bound. Please see also Reviewer 1, Comment 2.

7. Supplementary Figure 5: The map sigma or threshold for each panel needs to be assigned, and the nucleotide density cannot be cut away or isolated from the rest of the map. The entire map should be shown at each position to give readers a realistic view into the nucleotide binding pockets. Threshold levels should be specified overall or for each panel if different because of local resolution.

We appreciate the suggestion provided by the reviewer. Following the reviewer's advice, we have made revisions to Supplementary Figure 6 (formerly 5) to address the mentioned points.

8. Substrate directionality (page 14, line 5-7): can the authors please clarify what direction the substrate is going in, and label in Figure 3 panel b & e? Given the resolution, the authors should also provide a supplementary figure showing N-C and C-N fitting of the poly-A chain in the substrate density to provide rationale for the fit.

The density of the substrate is unfortunately not sufficiently resolved to provide a clear rationale for the fit. We fitted the substrate with N-C directionality because all available structures of substrate engaged AAA ATPases incl. Cdc48, process the substrate via a right-handed spiral staircase with N-C

directionality (see references below). Similarly, to the other member of the type II ATPase family Cdc48 (Cooney *et al.* 2019, Twomey *et al.* 2019), Pex1/Pex6 displays a right-handed staircase and is suggested to unfold ubiquitinated substrates (Schwerter *et al.* 2018, Pedrosa *et al.* 2018 Pedrosa *et al.* 2023). Therefore, we also modeled the substrate with N-C directionality, but we clearly state in the manuscript that the directionality of the substrate is not resolved.

The density of the substrate is ambiguous and we therefore modeled a continuous poly-alanine polypeptide backbone of 9 amino acids with N→C directionality (Fig. 3b). This is consistent with the translocation direction of other substrate-engaged cryo-EM structures of type II ATPases processing ubiquitinated substrates^{43,44,46}, as suggested for Pex1/Pex6^{24,40,52}, and exhibiting the same characteristic right-handed spiral staircase arrangement of pore loops 1, as observed in our structure.

Pan, M. et al. Mechanistic insight into substrate processing and allosteric inhibition of human p97. Nature structural & molecular biology 28, 614–625; 10.1038/s41594-021-00617-2 (2021).

Cooney, I. et al. Structure of the Cdc48 segregase in the act of unfolding an authentic substrate. Science (New York, N.Y.) 365, 502–505; 10.1126/science.aax0486 (2019).

La Peña, A. H. de, Goodall, E. A., Gates, S. N., Lander, G. C. & Martin, A. Substrate-engaged 26S proteasome structures reveal mechanisms for ATP-hydrolysis-driven translocation. Science (New York, N.Y.) 362, eaav0725; 10.1126/science.aav0725 (2018).

Twomey, E. C. et al. Substrate processing by the Cdc48 ATPase complex is initiated by ubiquitin unfolding. Science (New York, N.Y.) 365, eaax1033; 10.1126/science.aax1033 (2019).

Wald, J. et al. Mechanism of AAA+ ATPase-mediated RuvAB-Holliday junction branch migration. Nature 609, 630–639; 10.1038/s41586-022-05121-1 (2022).

Lo, Y.-H. et al. Cryo-EM structure of the essential ribosome assembly AAA-ATPase Rix7. Nature communications 10, 513; 10.1038/s41467-019-08373-0 (2019).

Puchades, C. et al. Structure of the mitochondrial inner membrane AAA+ protease YME1 gives insight into substrate processing. Science (New York, N.Y.) 358, eaao0464; 10.1126/science.aao0464 (2017).

9. The authors need to show the cryo-EM density for all pore loops engaged around the substrate with the substrate density. This can either be integrated into Figure 3 or shown in a supplementary figure.

Thank you for the suggestion. We added Supplementary Figure 11.

10. Supplementary Fig 1, a: what are the other major bands in the gel lanes? Can these be substrate? Possible to mass-spec those bands to ID. Have the authors tried western blot (i.e., against Ub?) to give some suggestion of the substrate identity?

We thank the reviewer for this suggestion. Unfortunately, we were not able to identify the substrate using MS (please see Reviewer 1; comment 9). According to the reviewer's comment, we further performed western blotting against Ub on the purified complex. As a control for the anti- Ubiquitin antibody, we used in vitro polyubiquitinated Eos. As a further control, we used anti- Pex6 antibody. Western blotting did not show any ubiquitinated protein present in our purified sample. However, mass spectrometry proteomics had one hit of a ubiquitinated protein which may be under the detection limit of western blotting (Ubiquitin-60S ribosomal protein L40 OS=*Saccharomyces cerevisiae*

(strain ATCC 204508 / S288c) OX=559292 GN=RPL40B PE=1 SV=1)), however there is currently no further evidence that this might be a physiologically relevant substrate.

Rebuttal Figure 3. Western blot against Ub suggests that the purified sample does not contain an ubiquitinated substrate.

11. I applaud the authors for deposition in EMPIAR.

We thank the reviewer for their comment. Open access to data enables independent validation, reuse, and integration of structural information. We strongly support the EMPIAR initiative, and due to the clear benefits it brings to our field, we hope that EMPIAR depositions will become mandatory in the future.

12. Can the authors conceptualize their findings with the recent import paper (Gao...Rapoport, 2022) to give an overall view of peroxisome transport?

We appreciate the suggestion provided by the reviewer. However, we have concerns that establishing an overall view and concept of peroxisomal receptor translocation and recycling is not possible until we gain insights into the entire peroxisomal export/import machinery. We have now uncovered the structure of a crucial component, the Pex1/Pex6 AAA translocase, which plays a vital role in receptor recycling. Additionally, two receptor pores, namely the receptor import pore, involving the docking complex, and the ring finger complex, have been suggested being in close proximity to each other. The structure and mechanism of the import pore remain unclear. These complexes may also be involved in a megacomplex associated with Pex1/Pex6. Pex15, the membrane anchor of Pex1/6, is expected to interact for example with both the docking and RING finger complexes. Furthermore, it remains unclear whether the receptor fully enters the peroxisomal lumen, and the factors determining the direction of this process are not yet understood. Given the current stage of our research, we believe that discussing and summarizing the recent exciting results on the export/import machinery in more detail is beyond the scope of this manuscript.

Reviewer #3 (Remarks to the Author):

Rutterman et al present cryo-EM structures of the Pex1/Pex6 AAA+ complex bound to substrate, providing insight into the molecular mechanisms underlying its translocase activity. Overall, the substrate-bound ATPases and the core mechanistic implications of these structures are essentially identical to those previously described for a wide variety of AAA+ proteins. However, given the distinct domain organization of the Pex1/Pex6 heterohexamer, these findings constitute an interesting addition to the extensive body of evidence supporting highly conserved mechanisms for AAA+ proteins. Moreover, these results shed light into the molecular principles underlying Pex1/Pex6 specific functionality, particularly regarding the organization of its unique N domains and the intersubunit communication interactions within the Pex1/Pex6 heterodimers. Still, this study would require major revisions prior to publication in Nature Communications.

We thank the referee for the positive comments on our study and for insightful questions.

Major points:

1. The novelty of this study stems from the identification of unique features of Pex1/Pex6, particularly the electrostatic network between the N2 domains and the communication hub around the nucleotide binding pocket. Can the authors confirm the functional relevance of their structural findings? For example, mutate some of the key residues involved in these interactions and biochemically characterize the mutants (e.g. complex assembly, ATP hydrolysis or translocase activity)?

We thank the reviewer for their comment. The structures presented in this study offer unprecedented mechanistic insights into the unique structure and molecular mechanism of the heterohexameric ATPase complex. Given the wealth of new structural data we have gathered, we acknowledge that fully characterizing the functional relevance of all the structural findings and analyzing the effects of mutations of all key residues is an enormous task beyond the scope of this manuscript's revision timeframe.

It is important to note that Pex1/Pex6, the molecular motor of peroxisomal biogenesis, has been extensively studied biophysically and biochemically in the past (Birschmann et al. 2005; Saffian et al. 2012; Grimm et al. 2012; Tan et al. 2016; Pedrosa et al. 2018; Birschmann et al. 2003; Gardner et al. 2018; Blok et al. 2015; Gardner et al. 2015; Krause et al. 1994; Erdmann et al. 1991; Rosenkranz et al. 2006; Faber et al. 1998; Geisbrecht et al. 1998; Yu et al. 2022). A plethora of these variants/deletions are discussed in our manuscript in the context of our structural findings.

Our data provide for example a clear and comprehensive explanation for previous biochemical studies that have highlighted the critical role of the D1 ring and N-terminal domains in complex assembly (Ciniawsky et al., 2015), while emphasizing that the primary activity of the complex is centered in the D2 ring (Birschmann et al., 2004). Furthermore, our findings offer valuable insights into the molecular mechanisms underlying complex assembly revealing how nucleotide binding in the D1 ring promotes this process, while ATP hydrolysis specifically occurs in the D2 ring (Gardner et al., 2015).

Moreover, it is worth noting that approximately 65% of patients with Peroxisomal Biogenesis Disorders are affected by mutations in the human Pex1/Pex6 complex, which is homologous to the yeast complex we studied. Our structure provides a solid molecular basis for understanding the impact of these mutations and their effects on complex function. However, a detailed discussion of these specific residues here would well go beyond the scope of the present manuscript.

2. Is it possible that the twin seam state is an artifact of the hybrid mutant complex (Walker B (WB) Pex6 but wild type Pex1) used for structural analysis? WB mutations delay hydrolysis. In FigS. 5b, the E subunit (WB Pex6) remains bound to ATP (has not yet hydrolyzed), whereas the D subunit (wtPex1) is bound to ADP (already hydrolyzed). This apparent “skip” of the sequential hydrolysis cycle might be a consequence of delayed hydrolysis of WB Pex6, and very unlikely in the context of the wild type complex. Did the authors study complexes with both Pex1 and Pex6 carrying WB mutations? Or complexes where both were wild type? Did they observe the double seam state in either case?

We appreciate the reviewer's comment (see also reviewer 1 comment 8). We have considered this possibility and have thoroughly reviewed the previous structural data on Pex1/6. The low-resolution cryoEM structure of WT Pex1/Pex6 (both wild-type) is available in the EMDB (Blok et al., 2015), and we carefully analyzed the respective density. From this comparison, it is clear that the WT structure (ATP γ S) adopts the twin seam state (see Rebuttal Figure 2; Reviewer 1; comment 8). This is characterized by the pronounced organization of the D2 ATPase ring of WT Pex1/Pex6 (ATP γ S) into three pairs of Pex1/Pex6, one of which is significantly tilted outwards (twin seam). In addition, the twin seam state was also observed in negative stain reconstructions of WT Pex1/Pex6 (Gardner et al., 2015).

Furthermore, we collected a small cryoEM dataset of the Pex1_WB/Pex6_WB complex (both Pex1 and Pex6 carrying WB mutations as suggested by the reviewer). 3D classification revealed again a similar twin-seam state as the major conformation (9.3 Å resolution; 51% (8969 particles) of the final 17490 particles set). (Rebuttal Figure 3b, gray reconstruction 3c).

Rebuttal Figure 4. Twin-seam state in Pex1_WB/Pex6_WB and Pex1/Pex6_WB variants a) Twin-seam state of the Pex1_WT/Pex6_WB (filtered to 9.3 Å resolution for direct comparison) and the new low resolution cryoEM structure of b) the Pex1_WB/Pex6_WB complex. c) a and b superimposed. The eam-subunit dimer is indicated by a black dashed line.

3. The resolution of the reconstructions and the quality of the derived atomic models are limited. While the resolution appears to be sufficient to support the authors' claims, further processing might help improve the resolution, which seems to be limited by unresolved sample heterogeneity and rigid body rotations of different parts of the complex relative to one another. In SFig. 2, it appears that the authors performed symmetry expansion. Did they ever do focused classification using a mask for a single heterodimer on the expanded dataset? Also, did they try masking individual rings or domains and doing focused classification on those? Or doing multibody refinement? Have the authors tried clustering the particles (3D Classification without alignment)? Or 3d variability analysis in cryosparc?

We have now significantly improved the quality of the molecular models (see Reviewer 1; comment 1). Furthermore, we would like to emphasize that we have strictly avoided any over-interpretation throughout the manuscript. We agree that the resolution of both cryoEM maps has a gradient with

well resolved D1 and D2 domains and lower but still interpretable resolved N-terminal domains and seam subunits. We agree with the reviewer that the resolution of these lower resolved areas is limited due to the flexibility and thank the reviewer for the suggestions to improve the resolution. Indeed, we have attempted focused classification and refinement of the extended dataset in Relion, but have not seen any significant improvement that would allow us to further resolve sample heterogeneity.

Following the reviewer's suggestion, we tried following approaches in cryosparc to improve the resolution:

- We used Cryosparc to perform a non-uniform refinement on both classes. While the nominal resolution of class 3 did not improve over the Relion map, class 4 improved from 4.7 to 4.4 Å with a tight automatically generated mask. However, visual analysis of the new density suggested an improvement in the N-terminal domains but overall less details in the higher resolution regions. Furthermore, further density modification of the respective half maps using Phenix ultimately resulted in the same resolutions and we see no differences in the respective maps. Thus, non-uniform refinement did not improve the quality of the volumes used for molecular modelling.

- We also tried the recently released 3D flexible refinement in Cryosparc which was able to resolve the movements between both conformations. Unfortunately, this reduced the nominal resolution of the consensus map to 4.27 Å.

- We also used cryosparc to perform a 3D classification after symmetry expansion of the best particles, focusing separately on the D1 ring, the D2 ring (excluding the seam subunit) and the dimers. The 3D classes were then refined locally using the focused mask. Despite this effort, the individual densities did not improve over the maps processed in Relion.

4. The authors describe the complex as a "trimer-of-dimers" and seem to conclude that this is due to the heterodimeric nature of the Pex1/Pex6 complex. However, a similar organization was also observed in Lon AAA+ protease, a homohexameric protein (Shin et al Nat. Comm 2021). Like in Pex1/Pex6, the N-domains of Lon appear to form a trimer of dimers and the substrate-bound Lon ATPases were found in a 4 ATP, 2 ADP state, despite being identical. Is it possible that N domain driven interactions might give rise to trimer-of-dimers organizations across different AAA+ proteins independent of homo- or heterohexamer?

In fact, in LoN proteases, the N-terminal domains form rigid coiled-coil homodimers that are tightly bound to the homohexameric ATPase ring, resulting in a "trimer-of-dimer" arrangement. Truncation of the NTDs in LoN results in loss of enzymatic activity. CryoEM structures of substrate-bound LoN also show two seam subunits in the ADP state. The NTD dimers are thought to act as a rigid extension of the ATPase ring and may be involved in repositioning the ATPase domains during the translocation cycle, triggering the displacement of two seam subunits through steric clashes.

At first sight, the organization is similar to that of Pex1/Pex6. However, the 'tight' NTD dimers in Pex1/Pex6 are heterodimers formed due to complementary charges between the domains of two different proteins. A characteristic "trimer of heterodimers" arrangement continues in the catalytically inactive heterohexamer in the D1 ring, which is absent in LoN. Thus, although the NTD heterodimers are critical for this organization, the architecture of this 'rigid extension' of the catalytically active D2 ring in Pex1/Pex6 is certainly more complex, as it is organized in two layers (NTDs and D1) with dimers formed due to distinct features and interfaces between two different proteins. Furthermore, whereas in LoN steric clashes lead to the disengagement of two seam subunits, in Pex1/Pex6 we observe the disengagement of a more compact "twin-seam" dimer with an ATP in the binding pocket between the seam subunits and the pore loops almost in the same plane. Pex1/6 thus has an additional level of

complexity. Rigid body motions associated with nucleotide exchange and ATP hydrolysis in the D2 ring are first amplified by the heterodimers of the D1 ring. We can however conclude in general that ATPase rings are often associated with rather rigid extensions that, depending on their organization (e.g. trimer of dimers, as in Pex1/Pex6 or LoN, or other scaffolds, e.g. ClpXP), result in adaptations of the otherwise conserved hand-over-hand mechanism of substrate translocation.

Pex1/Pex6 has thus developed unique structural features involving two distinct proteins organised in pairs in three layers to achieve its function, but there are indeed characteristic similarities to LoN, which we have now included in the discussion of the revised manuscript. We thank the reviewer for this comment!

At first glance, the organization resembles that of LoN proteases, in which NTD dimers crown a homohexameric ATPase ring, resulting in a "trimer-of-dimers" arrangement^{61,72,73}. However, the NTD dimers in Pex1/Pex6 are heterodimers formed by complementary charges between the domains of two different proteins (Fig. 2d,e). A characteristic "trimer of heterodimers" arrangement continues in the catalytically inactive heterohexamer in the D1 ring, which is absent in LoN proteases. While in LoN proteases steric conflicts lead to disentanglement of two seam subunits^{61,72,73}, in Pex1/Pex6 we observe disentanglement of a more compact "twin seam" dimer in the D2 ring with an ATP in the binding pocket between the seam subunits and the pore loops almost in the same plane (Fig. 4a,e). Pex1/Pex6 has thus developed unique structural features involving two distinct proteins organized in pairs in three layers to achieve its function, resulting in an adaptation of the otherwise conserved hand-over-hand mechanism.

5. I would recommend that the authors better separate their structural observations (results) from their conclusions (discussion). Throughout the manuscript, they propose some great interpretations, but I would encourage them to remove them from the results section and integrate them into an extended discussion regarding the implications of the unique structural features they identify in Pex1/Pex6. For example, the authors speculate in the results section that the position of the N domains might be nucleotide independent and cite previous studies. This would make sense to me, because the N2 domains sit atop the planar hydrolysis-incompetent D1 ring. However, I don't think the authors have enough evidence to support this statement in the results section, because they do not have any apo (nucleotide-free) subunits in their substrate-bound structures. Could the position of the N domains be the same in ADP and ATP states but different in an apo state? Can the D1 domains bind ADP or do they only ever bind ATP? Does a network of residues allosterically connect the N-domains to the nucleotide binding pocket (either D1 or D2) as seen in many other AAA+ proteins?

We thank the reviewer for this comment and have moved this interpretation to the Discussion section. Previous structural work (Ciniawsky et al. 2015, Gardner et al. 2015) using either ATP or ADP shows that the complex is able to assemble in the presence of either ADP/ATP, but disassembles in the absence of nucleotides (Birschmann et al. 2006, Saffian et al. 2012). The corresponding low resolution structures do not show any major differences in the N domains. Thus, we can conclude that the D1 ring can bind both ATP and ADP with almost no changes in the arrangement of the N-terminal domains (Ciniawsky et al. 2015, Gardner et al. 2015).

For Pex1/Pex6 we do not expect major conformational changes between ADP/ATP and the apo state, e.g. similar to p97, as we did not see any 2d or 3d classes with changes in the N-terminal domains, besides the described swing-in/out. Furthermore, our structures show the same arrangement as in the substrate-free structures.

There is indeed a long flexible linker flanking helix $\alpha 28$ of Pex1D2 with the Pex6N1 domain as described in Supplementary Figure 10c, which could be involved in the described swing-in/out mechanism, but we do not observe a dramatic conformational change similar to cdc48 (up and down conformation of the NTDs depending on the nucleotide state).

We agree with the reviewer that we are missing a clear apo state in our structural models and we cannot rule out such an extreme case. However, our data together with previous low resolution structures of ATP/ADP Pex1/6 support our conclusion about the relative rigidity of the N-terminal domains compared to other AAA ATPases, which we believe is important to acknowledge.

Minor points:

6. The authors mention interesting differences in ATP hydrolysis rates of the Pex1 vs Pex6 WB mutants. Could they add those results as a panel in a supplementary figure? Were the results consistent across diverse ATP concentrations?

We thank the reviewer for this question. We apologize for making it sound as if we had performed ATPase assays on these mutants. Instead, in the previous version of the manuscript, we cited previous studies that performed these assays and compared Pex1 vs. Pex6 WB expressed in bacteria in detail. These assays have been done across diverse ATP concentrations. To avoid confusion, we have separated our results and discussion more carefully. In addition, we have now performed an HPLC-based ATPase assay to investigate the residual ATPase activity of the Pex1/Pex6_WB variant. For details please see the response to reviewer 1, comment 4 and new supplementary figure 1.

7. The authors picked less than 2 million particles out of a huge dataset of 16000 micrographs. In the raw micrograph shown in FigS.1, many particles appear to be somewhat “disintegrated”. Can the authors comment on this? Did they observe different sample behavior when using different grids or freezing conditions?

In fact, we extensively tested various conditions before collecting such a large data set. The main reason for collecting such a large number of micrographs was to obtain a sufficient number of side views due to the preferred orientation of the particles. We have shown a random micrograph in Fig. S1 and we cannot rule out the possibility that the ice of this particular hole might have been too thin. We have now updated supplementary Fig. 2c to show a representative micrograph for the data set. Immediately after collecting the dataset, we carefully checked the 2D class averages, which were of high quality, so there is no problem with particle dissociation (see Supplementary Figure 2d).

8. I agree with the authors’ conclusion that the single and double seam states are substrate-bound states within the substrate translocation cycle. To prevent confusion, I would recommend not to call it an “open” state in the discussion, as this is often used to refer to the substrate disengaged state (sometimes also called “off” state) with an open seam observed for other AAA+ proteins in the absence of substrate (e.g Hsp104 (Gates et al Science 2017) and Lon).

We revised the manuscript, as per the reviewer’s suggestion.

Reviewer #4 (Remarks to the Author):

Heterohexameric AAA+ ATPase Pex1/Pex6 complex is important for peroxisome biogenesis. Both of Pex1 and Pex6 are composed of tandem N1 and N2 domain followed by two ATPase domains (D1 and

D2 domain). The D1 and D2 domains within Pex1/Pex6 can form two stacked hexameric rings, but only D2 domains are able to hydrolyze ATP. In this manuscript Rüttermann et al introduced Walker B mutation within the D2 domain of yeast Pex6 which reduced ATP hydrolysis while preserved the substrate binding, in beneficial of capturing translocation process. They further determined the cryo-EM structure of Pex1/Pex6 complex with an endogenous substrate bound in the D2 ring. From the cryo-EM data analysis, they analyzed two distinct conformations, ie single seam state Class 3 and twin seam state Class 4, and proposed a mechanism about how the Pex1/Pex6 functions to translocate the substrates in a right-handed spiral staircase manner via pore-1 loops. Overall, the manuscript appears to be appropriate interpretation on the structures with extensive and detailed discussion.

We thank the referee for the positive comments on our study and for insightful questions.

Specific comments:

1.The author focused on characterizing the better resolved Class 3 and Class4. However, there are 6 classes in total from 3D classification. Have the authors compared the details of the other 4 classes with Class 3 and Class4. Is it possible to differentiate more intermediate steps?

Thank you for this comment. We have indeed refined these other four classes and compared them in detail with classes 3 and 4. However, classes 1, 5 and 6 looked disaggregated with missing N-terminal domains, strong preferred orientation or missing core alpha helices. It was not possible to refine these classes below 6-8 Å. Class 2 is almost identical to class 3, but the overall resolution is lower. Combining the two classes into one and refining it further did not improve the resolution. We therefore focused our study on Class 3 and Class 4. Furthermore, we tried to resolve more variability as suggested by reviewer 3 (seem reviewer 3 comment 3) but did not improve the overall resolution.

2.The structures in this paper suggested that ATP within D1 domains plays a role in stabilization the Pex1 D1 small ATPase domain rather than inter-molecular association. Additional biochemical assay would strengthen the claims.

Indeed, the ATP bound to the Pex1 D1 domain does not contact the neighbouring Pex6 D1 domain, but biochemical assays performed by Krause *et al.*, 1994 and Gardner *et al.*, 2015 report that a mutation of the conserved Walker-A lysine in the D1 domain of Pex1 results on an assembly defect. We therefore agree with the reviewer and conclude that the ATP within the D1 domain of Pex1 plays a role in stabilization of the small ATPase domain of Pex1 rather in direct inter-molecular association. We therefore rephrased the sentence based on our data and those of previous studies:

Thus, ATP does not directly contribute to the interdimeric contact, but previous studies report that mutation of the conserved Walker A lysine 467 within the D1 domain of Pex1 results in an assembly defect^{37,57}. This suggests that although ATP at Pex1-D1 does not directly link the adjacent subunits, it may instead stabilize the small ATPase domain of Pex1, which is directly involved in the interdimer interaction.

3.The author showed that Pex1 (N2) and Pex2 (N2) can stabilize with each other via complementary electrostatic interaction. Could mutations in this interface abolished the assembly or activity of the complex?

Indeed, Ciniawsky et al. 2015 report that Pex6 N2-terminal deletions result in unstable protein expression or compromised hexamerization.

4. Please define the meaning of saturating concentration of ATP in page 6?

The ATP concentration that cannot be depleted within the timeframe of the purification to avoid any disassembly of the complex. During the purification 3mM ATP was used and additional 1mM of ATP was added after size-exclusion chromatography prior cryoEM preparation. Please see the materials and methods section for details. This is also defined in the main manuscript now.

5. In Supplementary Figure 1, the SDS-PAGE results showed besides of Pex1/Pex6, there are other obvious protein bands, what are the identity of these proteins? Are they potential substrates that were trapped in the pore?

Thank you for this comment. Please see our response to Reviewer 1, comment 9.

6. The protein expression and purification section in Supplementary part, there are two tandem paragraphs described pooling the elutes and running gel filtration, one of them may be redundant.

We have rephrased the respective sentence:

The concentrated protein fractions were separated by size on a Superose 6 10/300 column operated with an AEKTA Purifier (GE Healthcare Bioscience, Chicago, USA). UV-absorption was monitored and Pex1/Pex6(E832Q) fractions were pooled and concentrated to a final protein concentration of 3.6 mg/ml.

7. The authors should include B factors for protein and ligand in Supplementary Table 1.

Thank you, we have added the B factors for protein and ligand into supplementary table 1. Furthermore, we have improved the models significantly (see our response to Reviewer 1, Comment 1).

Before:

Model refinement		
	Class 3 Single-'seam' conformation	Class 4 Twin-'seam' conformation
Peptide chains	7	7
Residues	5566	5565
Ligands	ATP: 10, ADP: 2, Mg ²⁺ : 10	ATP: 10, ADP: 2, Mg ²⁺ : 10
RMSD Bond length (Å)	0.004	0.003
RMSD Bond angles (°)	0.745	0.733
Ramachandran outliers (%)	0.05	0.16
Ramachandran allowed (%)	9.47	13.49
Ramachandran favored (%)	90.47	86.34
Rotamer outliers (%)	0.04	0.02
MolProbity score	2.15	2.31
Clash score	12.53	14.9
EMRinger score	1.53	0.95
B-factors mean (Protein)	155.34	172.00

B-factors mean (Ligand)	93.89	127.56
-------	--------

Now:

Model refinement		
	Class 3 Single-‘seam’ conformation	Class 4 Twin-‘seam’ conformation
Peptide chains	7	7
Residues	5564	5564
Ligands	ATP: 10, ADP: 2, Mg ²⁺ : 10	ATP: 10, ADP: 2, Mg ²⁺ : 10
RMSD Bond length (Å)	0.003	0.004
RMSD Bond angles (°)	0.751	0.881
Ramachandran outliers (%)	0.04	0.00
Ramachandran allowed (%)	7.26	9.8
Ramachandran favored (%)	92.68	90.20
Rotamer outliers (%)	0.06	0.00
MolProbity score	1.81	1.89
Clash score	6.27	6.28
EMRinger score	1.53	0.99
B-factors mean (Protein)	132.26	131.73
B-factors mean (Ligand)	92.38	89.13

REVIEWERS' COMMENTS

Reviewer #1 (Remarks to the Author):

The revised manuscript has done an excellent job in responding to the comments I raised from the original submission. This is a solid paper that deepens our understanding of the diversity of AAA+ unfolding mechanisms. I have no further concerns.

Reviewer #2 (Remarks to the Author):

All concerns addressed.

Reviewer #3 (Remarks to the Author):

The authors have appropriately addressed all major comments and the revised manuscript is, in my opinion, appropriate for publication in nature communications

Reviewer #4 (Remarks to the Author):

I find the revised manuscript greatly improved. I also appreciate the time taken to address both my and the other reviewers' comments, therefore I am supportive of publication of this revised manuscript.